# Exploring inferred geomorphological sediment thickness as a new site proxy to predict ground-shaking amplification at regional scale. Application to Europe and Eastern Türkiye

Karina Loviknes[1,2], Fabrice Cotton[1,2], and Graeme Weatherill[1]

Helmholtz Centre Potsdam, GFZ German Research Centre for Geosciences 14467, Potsdam, Germany

University of Potsdam 14469, Potsdam, Germany

**Correspondence:** Karina Loviknes (karinalo@gfz-potsdam.de)

**Abstract.** To test whether a globally inferred sediment thickness value from geomorphological studies can be used as a proxy to predict earthquake site amplification, we derive site amplification models from the relation between empirical amplification for sites in Europe and Türkiye, and the geomorphological sediment thickness. The new site amplification predictions are then compared to predictions from site amplification models derived using the traditional site proxies, $V_{S30}$ inferred from slope, slope itself, as well as geological era and slope combined. The ability of each proxy to capture the site amplification is evaluated based on the reduction in site-to-site variability caused by each proxy. The results show that the highest reduction is caused by geological era and slope combined, while the geomorphological sediment thickness show a slightly larger or equal reduction in site-to-site variability as inferred $V_{S30}$ and slope. We therefore argue that including geology and geomorphology in site amplification modelling on regional scale can give an important added value and that globally or regionally inferred models for soil and sediment thickness from fields beyond engineering seismology can have a great potential in regional seismic hazard and risk assessments. Furthermore, the differences between the site amplification maps derived from different proxies capture the epistemic uncertainty of site amplification modelling. While, albeit the different proxies predict similar features on a large scale, local differences can be large. This shows that using only one proxy when predicting site amplification does not capture the full epistemic uncertainty, which is demonstrated by looking into detail on the site amplification maps predicted for Eastern Türkiye and Syria, where the devastating Kahramanmaraş Earthquake Sequence occurred in February 2023.

## 1 Introduction

Local geological features can have a strong impact on earthquake ground shaking. Especially at sites with mainly loose sediments, which have been observed to amplify the recorded ground motion. Knowing the soil and sediment composition of a site is therefore necessary for computing the possible earthquake site amplification for seismic hazard and risk assessments. For a single site and site-specific analysis, several site parameters for characterizing shallow site conditions (e.g., fundamental frequency f0, shear wave velocity profile, horizontal-to-vertical ratio HVSR, depth to bedrock etc.) can be obtained from seismic and geotechnical investigations and used to predict local site amplification (e.g., Bergamo et al., 2021; Cultrera et al., 2021; Trifunac, 2016; Derras et al., 2017). For larger areas and regional site-amplification analysis, however, the site conditions must

be derived from empirical relations between relevant proxies available through regional or global maps (e.g., Bergamo et al., 2022; Thompson et al., 2010; Weatherill et al., 2023). Currently, the common practice for characterizing site amplification in seismic hazard and risk assessment is using the average shear wave velocity of the upper 30 meters of the soil column ($V_{S30}$). For a single site the velocity profile and $V_{S30}$ can be measured directly, but for larger areas and regions, however, $V_{S30}$ must be inferred from other parameters. A much-used method to calculate $V_{S30}$ is using slope from digital elevation models (DEMs), following Wald and Allen (2007). This method is based on the hypothesis that steep (high) slopes generally have less sediments and therefore higher shear-wave velocity ($V_S$), while flat (low) slopes are more likely to be basins filled with sediments and thus with lower $V_S$. Wald and Allen (2007) used measured $V_{S30}$ to derive a relation between $V_{S30}$ and slope for active and stable tectonic regions separately and provided a global map of predicted $V_{S30}$ values. However, inferring $V_{S30}$ based on slope has several limitations. As already stated by Wald and Allen (2007), the assumption of correlation between $V_{S30}$ and slope breaks down for continental glaciated terrains and nominally flat volcanic plateaus. In addition, Lemoine et al. (2012) have shown that other geological conditions, in particular narrow sedimentary basins and small topographic heterogeneity, have a poor correlation with the $V_{S30}$ model based on slope.

Since the Wald and Allen (2007) model, several $V_{S30}$ maps based on new methods and other geological proxies in addition to slope has been made, both on local and national level (e.g., Thompson et al., 2014; Vilanova et al., 2018; Foster et al., 2019; Mori et al., 2020; Li et al., 2022). However, as also argued by Weatherill et al. (2020, 2023), the main purpose of $V_{S30}$ is as a proxy to predict site amplification, and when inferring $V_{S30}$ from other parameters, it thus becomes a proxy-of-a-proxy. In fact, site amplification predicted by $V_{S30}$ based on slope show little improvement to the site-amplification models based directly on slope (Weatherill et al., 2020). Furthermore, it is important to keep in mind that inferred $V_{S30}$ should not be used interchangeably with measured $V_{S30}$ values without properly accounting for the additional uncertainty related to the $V_{S30}$ calculations (Lemoine et al., 2012; Thompson and Wald, 2016; Weatherill et al., 2023). The variability of ground-motion predictions can have great impact on the resulting probabilistic seismic hazard and risk assessments. Properly accounting for and separating between aleatory uncertainty, coming from natural randomness, and epistemic uncertainty, due to lack of knowledge, is therefore important. It has long been acknowledged that the site-variability is a significant contributor to ground-motion variability (e.g., Atkinson, 2006; Rodriguez-Marek et al., 2013). In particularly, using inferred site proxies in place of measured site parameters results in an increase in uncertainty. To account for this increase in uncertainty, Weatherill et al. (2023) derived separate site-amplification models for measured and inferred proxies and compared their impact on the final hazard and risk calculation. It was found that, although the median amplification predicted using inferred $V_{S30}$ were notably lower than the median predicted amplification using measured $V_{S30}$, the resulting seismic hazard and risk curves from the different approaches were within the same range. This emphasizes how seismic hazard is not only controlled by the median amplification, but also by the uncertainty. Indeed this increase in uncertainty, related to inferred proxies, compensates for the change in predicted median amplification in a probabilistic hazard and risk context.

In this study, we follow the approach of Weatherill et al. (2020, 2023), to test the suitability of new site proxies as predictors for site amplification and investigate the effect on epistemic uncertainty when using different regional and globally available site proxies. We skip the step of deriving a site-proxy ourselves and look beyond the field of engineering seismology for already

available large-scale models of soil and sediment conditions that would allow for inference of soil amplification across a wide
region. One of such models is the Pelletier et al. (2016) geomorphological model for sedimentary thickness. As the thickness
of soil and sediments down to bedrock is an important factor for modelling amplification of earthquake ground shaking, the
thickness of porous weathered material above unweathered bedrock is necessary for land surface modelling of, for example,
the water and carbon cycle (Pelletier et al., 2016). Pelletier et al. (2016) therefore developed a global data set of soil, intact
regolith, and sedimentary deposit thicknesses intended as input for hydrology and ecosystem models. The model is based on a
combination of data including slope, lithology and stratigraphy, and water table depth, all of which correlate with geotechnical
soil conditions known to yield seismic amplification. Because it is based on more robust geomorphological theories than
traditional inferred site proxies, like $V_{S30}$ based on slope or geology, we acknowledge the potential value of the Pelletier et al.
(2016) model and other similar large-scale models of soil thickness derived from other fields than our own, as possible input
in site amplification modelling in large scale seismic hazard and risk modelling.

To test the whether the geomorphological model can provide extra information and are suitable for ground-shaking predic-
tion, we derive a simple site-amplification prediction model using site-to-site residuals ($\delta S2S_s$)from the European Engineering
Strong-Motion (ESM) dataset (Lanzano et al., 2019; Luzi et al., 2020). The site-to-site residuals ($\delta S2S_s$) are derived from
a simple ground-motion model (GMM) following the method of Kotha et al. (2018, 2020). We compare the ability of the
geomorphological model to predict site amplification, to site amplification models based on the traditional site proxies; $V_{S30}$
derived from slope from Wald and Allen (2007), and slope alone, as well as a combination of slope and geological era. To
better investigate the differences in the site-amplification prediction maps derived from the different proxies, we focus on East-
ern Türkiye and Syria where the recent February 2023 Kahramanmaraş Earthquake Sequence occurred (Melgar et al., 2023;
Petersen et al., 2023).

## 2 Site-to-site terms

The site response at sites where ground motion records are available is often derived as the standard spectral ratio (SSR)
between a site and a nearby rock reference, or, in the rare cases it is available, a borehole reference at the same site. However,
a nearby rock reference or borehole stations are not always available for ground motion recording stations. Instead, when
a station has recorded several earthquakes, the repeatable site response can be separated from the source and path effect of
the ground motion, using methods like generalized inversion technique (GIT, e.g., Nakano et al., 2015), empirical spectral
modelling technique (Edwards et al., 2013) or empirical ground motion modelling (e.g., Kotha et al., 2017, 2018; Ktenidou
et al., 2018). In this study we use the latter method to remove the source and path effect from the ground motions using a
simple GMM. To derive the GMM we use robust mixed-effects regression (rlmm, Koller, 2016), where statistical outliers are
down-weighted and hierarchical data is dealt with by distinguishing between fixed effects as explanatory variables and random
effects as grouping factors (Bates et al., 2015). A GMM is typically composed of three main explanatory variables describing
the source, path and site effects of the ground motion. In its most basic form, magnitude and distance are used to describe the

source and path, while $V_{S30}$ is usually used to characterize the site effects. Here, however, only the source and path effects are used as fixed effects, while the site is included as a random effect:

$$\ln(\mu) = f_{R,g}(R_{JB}) + f_{R,a}(R_{JB}) + f_M(M_W) + \delta B_e + \delta S2S_s + \delta WS_{e,s} \tag{1}$$

$$f_{R,g}(R_{JB}) = c_1 \ln \sqrt{\frac{(R_{JB}^2 + h_D^2)}{(R_{ref}^2 + h_D^2)}} \tag{2}$$

$$f_{R,a}(R_{JB}) = \frac{c_3}{100}(\sqrt{R_{JB}^2 + h_D^2} - \sqrt{R_{ref}^2 + h_D^2}) \tag{3}$$

$$f_M(M_W) = \begin{cases} b_1(M_W - M_h) + b_2(M_W - M_h)^2 & \text{if} \quad M_W \leq M_h \\ b_3(M_W - M_h) & \text{if} \quad M_h < M_W \end{cases} \tag{4}$$

where, $\ln(\mu)$ is the median ground motion prediction and $f_{R,g}(M_W, R_{JB})$, $f_{R,a}(R_{JB})$ and $f_M(M_W)$ are the fixed effects capturing the scaling of the ground motion with geometric spreading, anelastic attenuation and magnitude for Joyner-Boore distance $R_{JB}$, hypocentral depth $h_D$ and magnitude $M_W$. The reference values, $R_{ref} = 30\,km$, $M_h = 5.7$ and $h_D = 4$, 8 and

100 $12\,km$, are frequency-independent and defined by Kotha et al. (2022). Following the notation of Al Atik et al. (2010), the between-event random effect $\delta B_e$ and site-to-site random effect $\delta S2S_s$ represent the systematic deviation of recorded ground motions from the GMM median predictions related to an event $e$ and a site $s$, respectively, and $\delta WS_{e,s}$ is the "remaining" record-to-record variability (Kotha et al., 2018; Loviknes et al., 2021). If a site-proxy dependent site term were included in the fixed effect, $\delta S2S_s$ would represent the systematic deviation of the observed amplification at site s from the median

amplification predicted by the model using the site proxy (Al Atik et al., 2010). However, because no site-proxy dependent site term is included in the GMM derived here, $\delta S2S_s$ captures all the site-specific response and thus can be used as an empirical site-amplification function describing the local amplification, or deamplification, of each station with respect to the median of all sites (Kotha et al., 2018). $\delta S2S_s$ is comparable to site amplification from GIT, as shown by recent studies (Bindi et al., 2017; Wang et al., 2023). The $\delta S2S_s$ is assumed to follow a frequency-dependent normal distribution with standard deviation; $\phi_{s2s}$:

$$\delta S2S_s = \mathcal{N}(0, \phi_{s2s}) \tag{5}$$

In this study the GMM and corresponding $\delta S2S_s$, is derived in the Fourier Amplitude Spectra (FAS) from the ESM dataset (Lanzano et al., 2019; Luzi et al., 2020). While most GMMs are derived for response spectral amplitudes (SA), representing the damped response of an elastic single-degree-of-freedom oscillator, we here derive the GMM and $\delta S2S_s$ in FAS to better

capture the physical effects that can be masked in the response spectra, in particularly at high frequencies (Kotha et al., 2022;

Bora et al., 2019; Bayless and Abrahamson, 2019).

To derive the GMM we use the same data selection criteria and similar functional form as Kotha et al. (2022). The GMM of Kotha et al. (2022) is a regionally adaptable models in FAS for shallow crustal earthquakes in Europe and Mediterranean regions. The regionalization in these models is represented by including an earthquake locality-to-locality variability term and an attenuation region-to-region variability term to the random effects. In the GMM derived for this study, these terms are not

included and only event and site are used as random effects, this done to minimize the possibility that regional differences in site effects propagate into the region-to-region random-effect.

Unlike traditional site-amplification factors, $\delta S2S_s$ is not relative to a reference rock condition, but to $\delta S2S_s = 0$ which is the centre of the distribution, median, of all the stations. The final $\delta S2S_s$ dataset contains site terms in the frequency range $f = 0.460 - 9.903\,\text{Hz}$ for 1680 stations in Europe and the Middle East, as shown on the map in Fig. 1 at $f = 0.529$, $1.062$

and $9.903\,\text{Hz}$. Although the site amplification shows a high variability and are mainly dominated by very local effects, some regional effects can be observed, for example for Italy the amplification is mainly high (above the median, red) in the Po-Plains and low (below the median, blue) in the Alps.

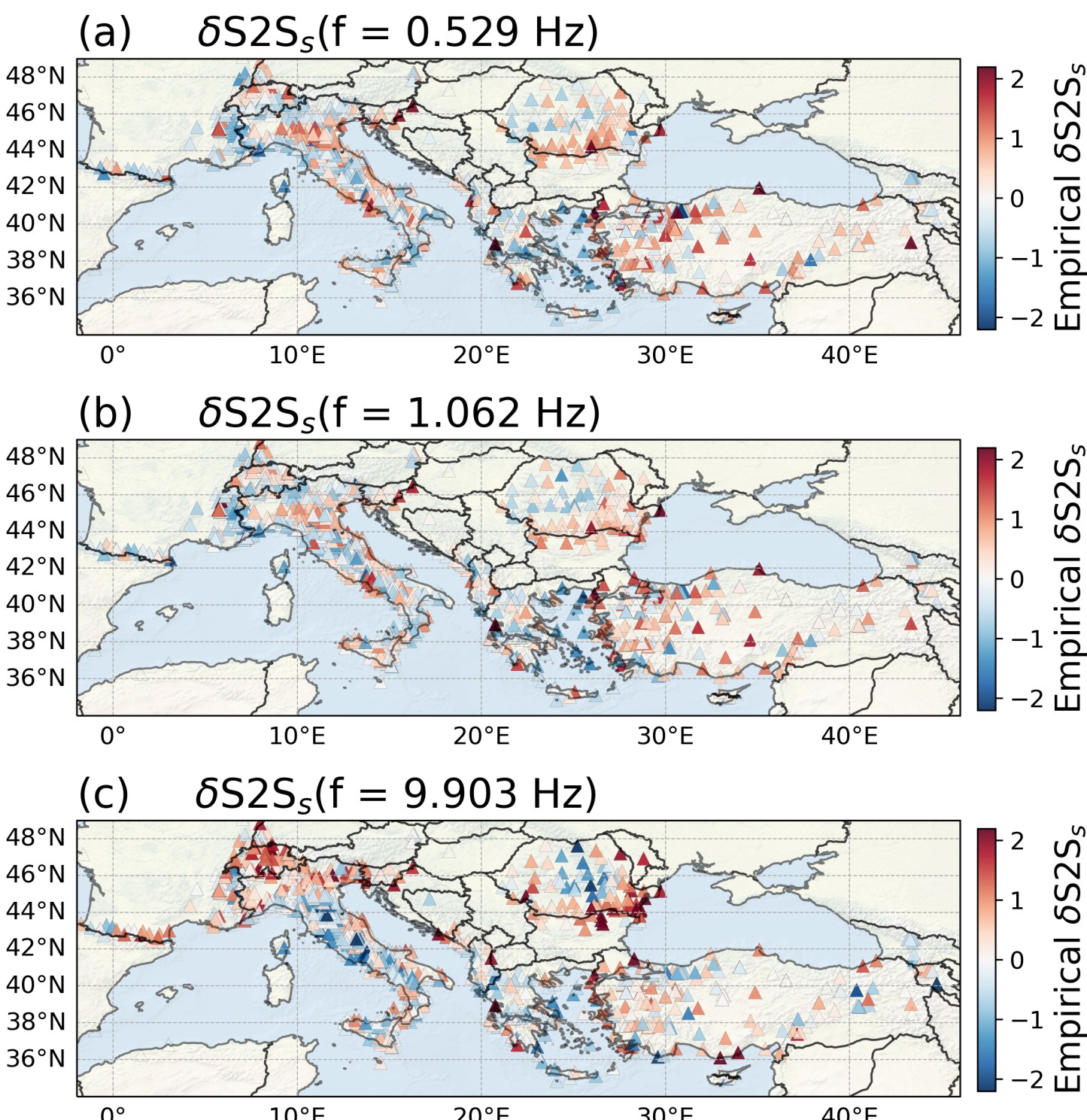

**Figure 1.** Map of the site-amplification factor $\delta S2S_s$ for (a) $f = 0.529\,\mathrm{Hz}$, (b) $f = 1.062\,\mathrm{Hz}$ and (c) $f = 9.903\,\mathrm{Hz}$. The colour scale shows the amplification for each station, where red represents amplified ground motions with respect to the median of all the stations, and blue represents deamplified ground motions.

# 3 Site proxies; inferred $V_{S30}$, slope, geomorphological sediment thickness and geological era

## 3.1 Inferred $V_{S30}$ from slope

The $V_{S30}$ dataset of Wald and Allen (2007) has had important implications for large-scale seismic hazard and risk assessments and is arguably the most used inferred site proxy in seismic hazard and risk studies (Silva et al., 2020). Wald and Allen (2007) used measured $V_{S30}$ from several location in United States, Taiwan, Italy and Australia to derive a relation between $V_{S30}$ and slope, separating between active and stable tectonic regions. The slope was calculated from global 30 arc sec DEMs from the Shuttle Radar Topography Mission (SRTM30). Here we use the inferred $V_{S30}$ values for Europe directly from the global map published by Wald and Allen (2007). These values range from $V_{S30} = 180 \, \mathrm{m/s}$ to $900 \, \mathrm{m/s}$ as shown in the map of Europe in Fig. 2a and the distribution plot in Fig. 3a.

## 3.2 Geomorphological Sediment Thickness (GST)

The Pelletier et al. (2016) model provides a gridded global dataset of soil, intact regolith, and sedimentary deposit thicknesses down to 50 meters. In their model, Pelletier et al. (2016) defines bedrock as the unweathered bedrock below unconsolidated
material, which in lowlands are mainly considered sedimentary deposits, and high porosity material, which in uplands can be divided into regolith and soil where soil is the material that sustain life and regolith is the porous weathered material below soil (Pelletier et al., 2016; Holbrook et al., 2014). The model is therefore developed by partitioning the Earth's surface into uplands and lowlands, which are then separated into hillslopes and valley bottoms. Uplands and lowlands are defined as areas undergoing net erosion and net deposition, respectively, over geological time scales and are distinguished using geological
maps and topographic analysis. Hillslopes and valley bottoms are identified using topographic curvature from DEMs, where hillslopes are areas of unconfined surface water flow, while valley bottoms are areas of confined surface water flow. This distinction is particularly important for uplands, and the regolith, soil and sediment thickness values are derived separately for the three landform types: upland hillslopes, upland valley bottoms and lowlands. The values are calculated using mathematical formulas specific to each landform, based on World climate data, water table depths, soil thickness databases and depth-to-
bedrock data, among others, as input. The final dataset provided by Pelletier et al. (2016) includes several 30 arcsec pixel grids covering $60°S - 90°N$ and $180°W - 180°E$, separating between maximum upland regolith, average soil thickness for upland hillslope, average soil and sediment thickness for upland valley bottom and lowlands, and average soil and sediment thickness across all areas. This study uses the grids for average soil and sediment thickness for all areas, from hereon referred to as geomorphological sedimentary thickness. The Pelletier et al. (2016) model has been previously tested as a proxy for basin
depth in Japan (Weatherill et al., 2020) and was included in the open-source site database of strong-motion stations in Japan by Zhu et al. (2021). However, both Weatherill et al. (2020) and Zhu et al. (2021) used the average soil and sediment thickness for upland valley bottom and lowlands grid, while here we use the average soil and sediment thickness for all areas, which is a combination of the grids for average soil thickness for upland hillslope and average soil and sediment thickness for upland valley bottom and lowlands, in order to access more values over a broader area. The geomorphological sedimentary thickness
ranges between 0 m and 50 m and is shown in the map of Europe in Fig. 2c and the distribution plot in Fig. 3c.

### 3.3 Geological era and slope for Europe

In the latest European Seismic Hazard and Risk model (ESHM20, ESRM20), geological era and slope are used to derive the site-response model (Crowley et al., 2021; Weatherill et al., 2023). This approach is based on Vilanova et al. (2018) who made a $V_{S30}$ map for Portugal from geological maps, and Weatherill et al. (2020) who compared several approaches for deriving site amplification from inferred proxies in Japan, including geology and slope. The harmonized surface geology map of Europe is a combination of three geological maps for Europe and Iceland and has a resolution of up to 1:1,500,000. Because several geological units, both following lithologic (nature) and stratigraphic (age) classification, contain too few stations, the geological units were grouped into the following seven geological eras: Holocene, Pleistocene, Cenozoic, Cretaceous, Jurassic-Triassic, Pre-Cambrian, and Paleozoic. The map and station distribution of these eras are shown in Fig. 2d and 2d. The slope used in this model was calculated from the 2014 General Bathymetric Chart of the Oceans grid (GEBCO 2014, https://www.gebco.net/) and is shown in Fig. 2b and 2b. In this study, we use the same slope and geological eras as Weatherill et al. (2023), with a resolution of 30 arc-seconds and 1:1,500,000, respectively, which are available on the EFEHR seismic risk web services (http://risk.efehr.org/site-model/).

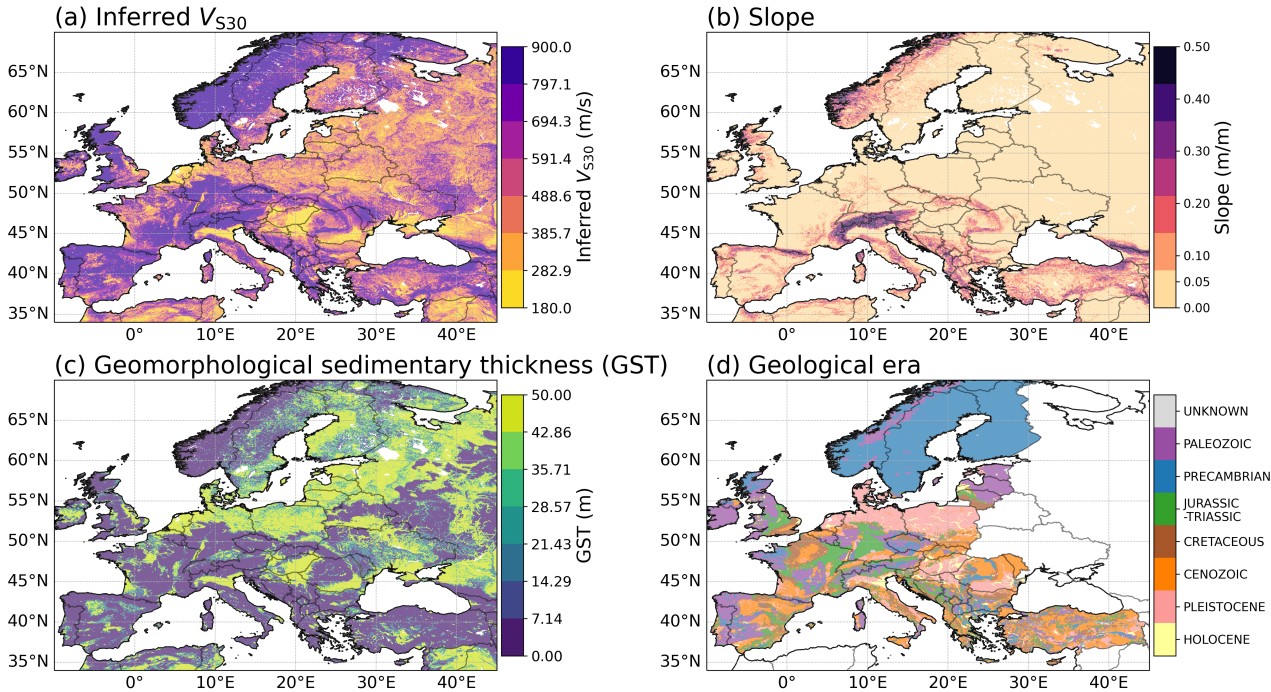

**Figure 2.** Map of the site proxies to be tested in this study. (a) $V_{S30}$ from slope by Wald and Allen (2007), (b) slope calculated from digital elevation models, (c) geomorphological sedimentary thickness by Pelletier et al. (2016), and (d) geological era used in the latest European Seismic Risk model (ESHM20, Crowley et al., 2021; Weatherill et al., 2023).

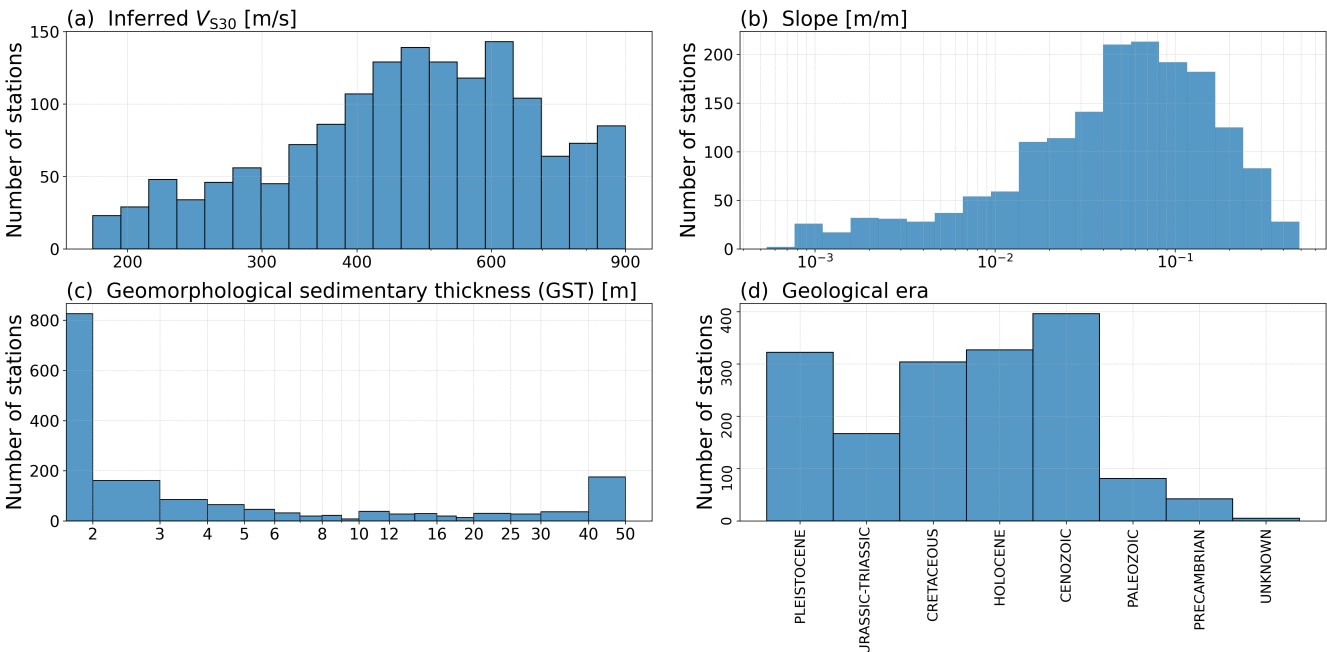

**Figure 3.** The distribution of the site proxies to be tested in this study. (a) $V_{S30}$ from slope by Wald and Allen (2007), (b) slope calculated from digital elevation models, (c) geomorphological sedimentary thickness by Pelletier et al. (2016), and (d) geological era used in the latest European Seismic Risk model (ESHM20, Crowley et al., 2021; Weatherill et al., 2023).

## 4 Amplifications predictions according to the different proxies

We evaluate the ability of the different proxies to predict site amplification by deriving a site-amplification model for each proxy using linear regression to capture the relation between the empirical amplification $\delta S2S_s(f)$ and the proxies:

$$Y_s(f, Proxy) = a\ln(x_{Proxy}) + b \tag{6}$$

where $Y_s(f, Proxy)$ is the predicted site amplification for a site $s$ at frequency $f$ using a proxy $x_{Proxy}$, and $a$ and $b$ are the coefficients derived from the linear regression.

A log-normal distribution is generally assumed for $V_{S30}$ and depth to bedrock (e.g., Boore et al., 2011; Vilanova et al., 2018), and the relation between site amplification and measured $V_{S30}$ is often derived as log-linear (e.g., Seyhan and Stewart, 2014; Derras et al., 2017). Indeed, Figure 4 shows the linear regression between the empirical amplification $\delta S2S_s(f)$ at frequencies $f = 0.529$ and $1.062\,\text{Hz}$, with measured $V_{S30}$. The coefficient of determination $r^2$ shows that the correlation between $\delta S2S_s(f)$ and $\ln(V_{S30})$ is higher than between $\delta S2S_s(f)$ and $V_{S30}$ in linear scale. We therefore, and following common practise, also assume a log-linear relation between the site amplification and the inferred site proxies.

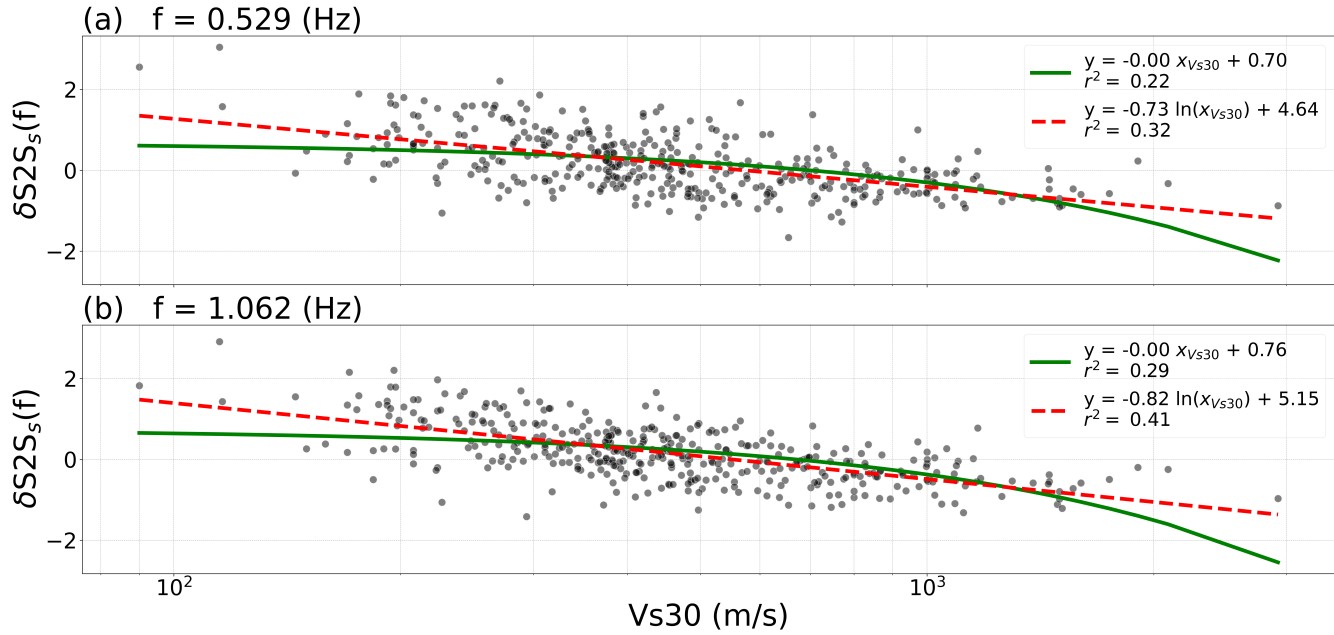

**Figure 4.** The linear (green line) and log-linear (red dotted line) relation between $\delta S2S_s$ and measured $V_{S30}$ for the frequencies (a) $f = 0.529\,\text{Hz}$ and (b) $f = 1.062\,\text{Hz}$.

As shown by the distribution of the inferred proxies (Fig. 3), the log-normal assumption is not fully fulfilled for inferred $V_{S30}$ and especially geomorphological sediment thickness. This is because the proxies are limited to a certain range during their calculation process. In the case of geomorphological sediment thickness, Pelletier et al. (2016) set the maximum value to 50 m, effectively meaning the thickness of the sediment layer is 50 m or more. Likewise, the inferred $V_{S30}$ is limited to a maximum value of 900 m/s.

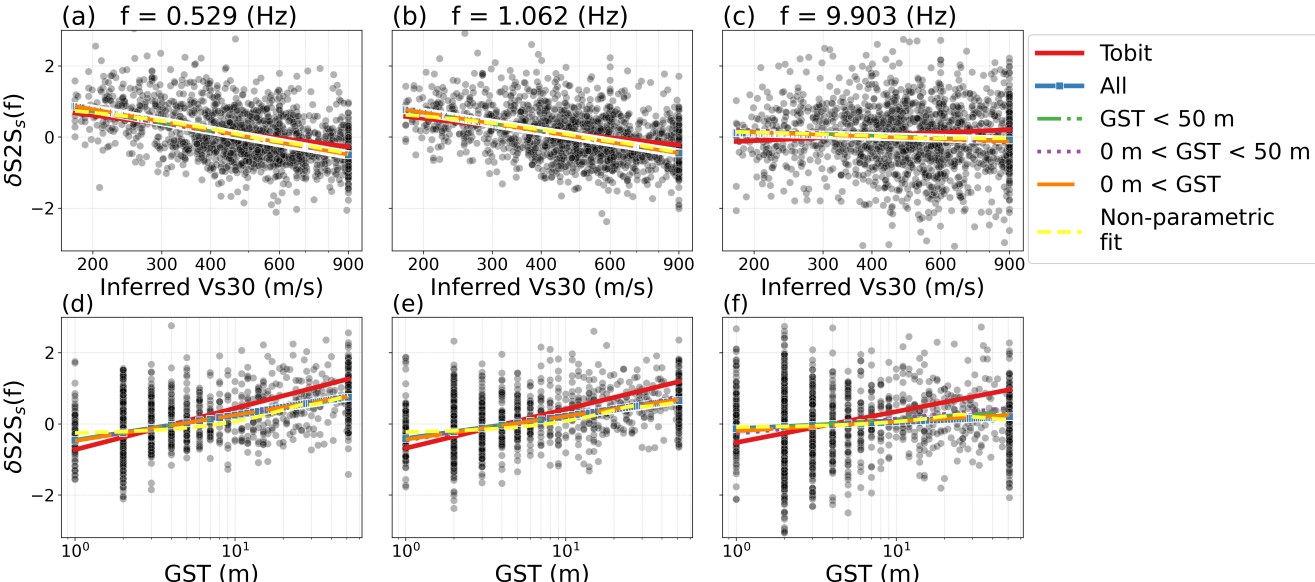

**Figure 5.** Inferred $V_{\text{S30}}$ (a, b, c) and geomorphological sedimentary thickness (d, e, f) with the $\delta\text{S2S}_s(f)$ of all the 1680 stations (black dots) in the ESM dataset for the frequencies $f = 0.529\,\text{Hz}$, (a, d), $f = 1.062\,\text{Hz}$ (b, e) and $f = 9.903\,\text{Hz}$ (c, f). The regression lines are from the Tobit regression (solid red line), and the linear regression on all the data ( blue line), on the selected dataset without the maximum geomorphological sediment thickness values (50 m, green line), without the extreme values of geomorphological sediment thickness (0 and 50 m, dotted purple line), and the minimum geomorphological sediment thickness (0 m, orange line). The general trend of the data is shown by a non-parametric fit (dashed yellow line).

When dealing with such uneven distribution caused by censoring the data, so-called "censored data", Tobit regression (Tobin, 1958) is a possibility. The Tobit model is developed to estimate linear relationships when the dependent variable is censored and uses the likelihood function to deal with the uneven distribution (Amemiya, 1984). However, as can be seen in Fig. 5 (red line), the Tobit regression strongly overestimates the slope of the relation between the site proxies and site amplification, which is also demonstrated by the non-parametric fit (dashed yellow lines, Fig. 5) and the low coefficient of determination $r^2$ for the Tobit regression (red line Fig. 6) compared to the other regression models, as shown in Figure 6. The slope from the Tobit regression is especially overestimated at high frequencies, where a weak relation between the empirical site amplification and site proxies is expected due to the impact of small-scale heterogeneities at the site where even the location or housing of the strong-motion station can affect the amplification (Hollender et al., 2020) and the coarse spatial resolution (30 arc second) of the site amplification model. Interestingly, at very low frequencies (below 0.5 Hz), the coefficients of determination $r^2$ for the regressions based on geomorphological sediment thickness are slightly reduced. This behaviour is not observed for $r^2$ based on the regressions with inferred $V_{\text{S30}}$ and might be caused by the geomorphological sediment thickness being limited to 50 m depth.

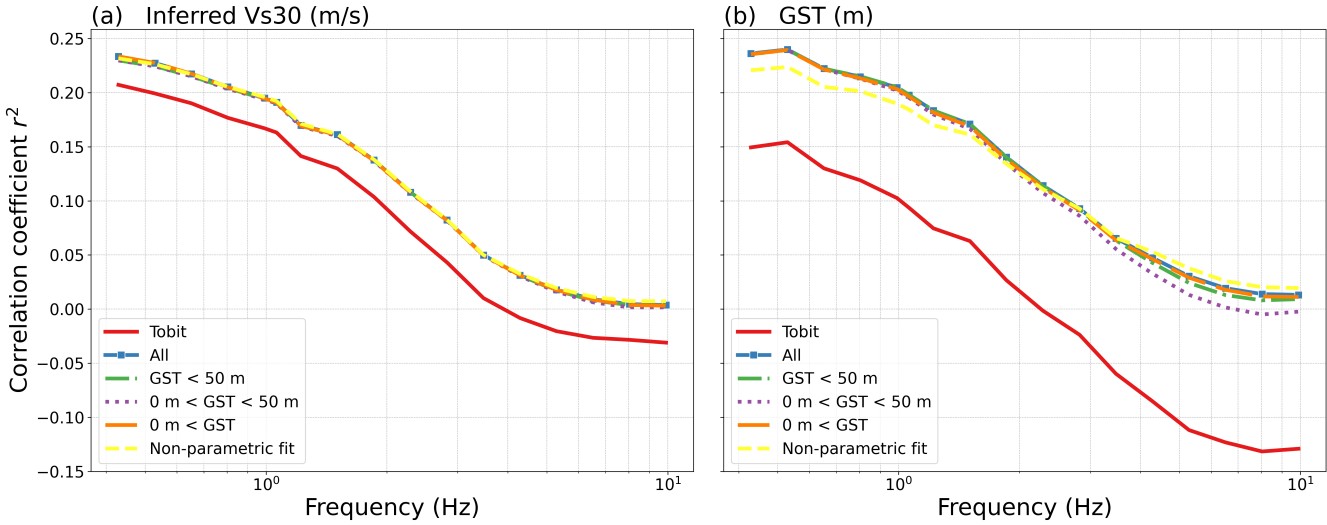

**Figure 6.** Coefficient of determination $r^2$ for the regressions shown in Fig. 5 between $\delta S2S_s(f)$ and the site proxies (a) inferred $V_{S30}$ and (b) geomorphological sedimentary thickness.

Another alternative approach to deal with the censored data is to exclude the end values when running the regression, however, this step only excludes a high number of sites while not having a strong impact on the regression line (Fig. 5, green, purple, orange and yellow line). Instead, we only omit sites with geomorphological sediment thickness = 0 m, which is the value that causes the highest unevenness in the distribution, as well as sites with missing values for any of the proxies, leaving us with 1508 sites for the regression. To evaluate the dependency of the regression on the selection of sites, we run a 10-fold cross-validation test, which is a method to separate the data used for the regression (training set) and the data used to validate the regression (validation set) when dealing with small datasets (Bishop and Nasrabadi, 2006). Because our dataset consists of 1508 sites, we choose to separate the data into 10 parts in order to have a sufficiently large validation set (about 150 sites). In the 10-fold cross-validation test, the dataset is thus split into 10 equal parts and the model is derived on 10-1 parts and tested on the remaining 1 part of the dataset. This is done 10 times and for each run, a different subset of the data is used for validation. The distributions of the site proxies for each of the 10 cross-validation iterations are shown in Fig. A1 in the Appendix.

Because the object of this study is not to find the best possible relation between each proxy and empirical site amplification, but rather to compare the ability of the proxies to predict site amplifications, we choose linear regression for simplicity. We do, however, acknowledge that site amplification is a complex phenomenon that the linear assumption cannot fully capture. Furthermore, because the models are based on the proxies' relation with $\delta S2S_s$ the resulting amplification predictions are, as $\delta S2S_s$, relative to the median prediction of the associated GMM used to obtain the $\delta S2S_s$ Eq. (1 - 4).

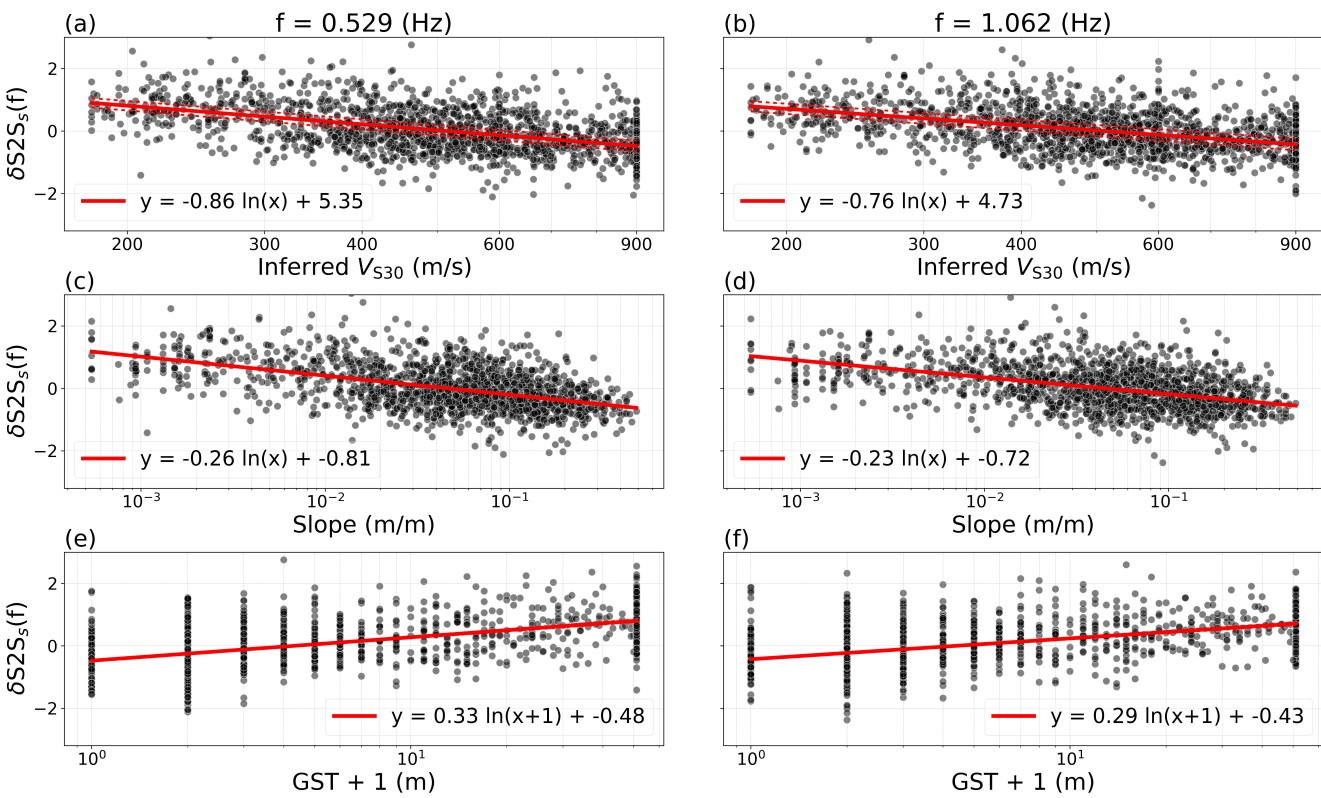

**Figure 7.** Linear regression (red lines) over the site proxies inferred $V_{S30}$ (a, b), slope (c, d) and geomorphological sedimentary thickness (e, f), with the station $\delta S2S_s$ (black dots) for the frequencies $f = 0.529\,\text{Hz}$, (left column; a, c, e) and $f = 1.062\,\text{Hz}$ (right column; b, d, f).

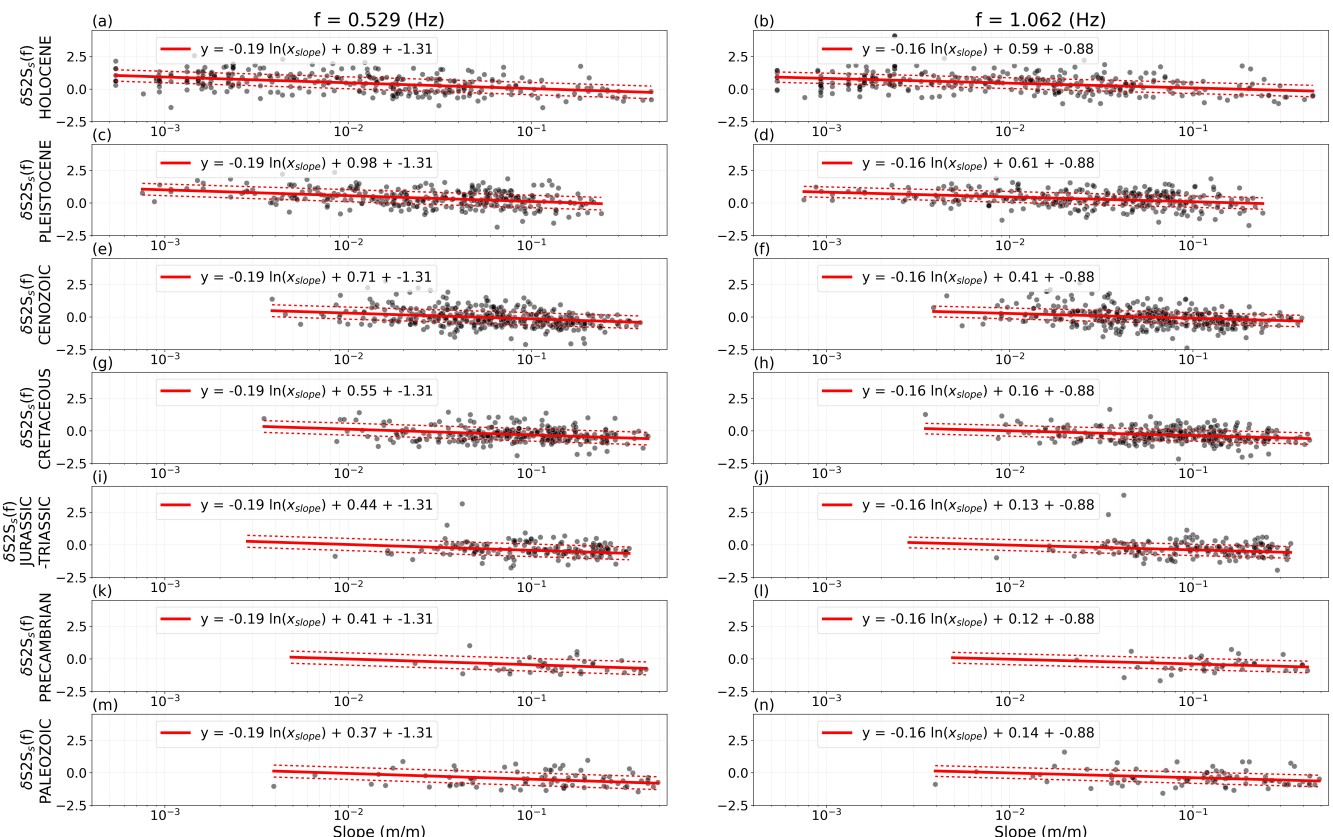

**Figure 8.** Linear regression (red lines) over slope and the geological eras; Holocene (a, b), Pleistocene (c, d), Cenozoic (e, f), Cretaceous (g, h), Jurassic-Triassic (i, j), Precambrian (k, l) and Palaeozoic (m, n), with the station $\delta S2S_s$ (black dots) for the frequencies $f = 0.529\,\mathrm{Hz}$, (left) and $f = 1.062\,\mathrm{Hz}$ (right).

We derive site-amplification models from linear regression between $\delta S2S_s$ for the frequency range $f = 0.460 - 9.903\,\mathrm{Hz}$ and the site proxies, $V_{S30}$ from slope, slope alone, geomorphological sediment thickness and geological era combined with slope. The selected sites and the regression lines are shown for $f = 0.529\,\mathrm{Hz}$ and $f = 1.062\,\mathrm{Hz}$ in Fig. 7 for inferred $V_{S30}$, slope and geomorphological sediment thickness. Although the $\delta S2S_s$ shows a high scatter with the site proxies, a general trend of higher amplification for low $V_{S30}$, low slope and high sediment thickness and low amplification for vice versa, can be identified. For

the regression over geological era and slope combined, we apply multiple linear regression, meaning with multiple independent variables where the categorical predictor, geological era, is transformed to dummy variables for each era and the regression then derives a constant coefficient for slope with a different intercept for each geological era (Fig. 8). The variation in prediction coefficients due to the alteration of the training-set in the cross-validation process, shown as dotted red lines in Fig. 7 and 8, is small for all the proxies except inferred $V_{S30}$ and geological era. This indicates that the site amplification models based on

inferred $V_{S30}$ and geological era and slope combined are more dependent on the data selection than the other proxies, causing

a higher final uncertainty. The coefficient of determination for the linear regressions shown in Fig. 7 and 8 and for the entire frequency range are shown in Fig. A2 in the Appendix.

## 5   Reduction in site-to-site variability

After deriving an amplification model based on each proxy, we compare the predicted amplification $\delta S2S_s(f, Proxy)$ with the empirical amplification $\delta S2S_s(f)$ at a site $s$ and frequency $f$. We measure the ability of each proxy to capture the site amplification using the reduction in site-to-site variability $\phi_{s2s}$ as an indicator of the efficiency of each proxy in predicting the amplification Stewart et al. (2017); Zhu et al. (2022). We compute the corrected site term for each proxy-specific predicted amplification $\delta S2S_{s,cor.}(f, Proxy)$:

$$\delta S2S_{s,cor.}(f, Proxy) = \delta S2S(f) - \delta S2S_s(f, Proxy) \tag{7}$$

$$\delta S2S_{s,corr.}(f, Proxy) = \mathcal{N}(0, \phi_{s2s_{cor.}}(f, Proxy)) \tag{8}$$

Where $\delta S2S_{s,cor.}(f, Proxy)$ represents the remaining site amplification that is not captured by the proxy-based amplification prediction $\delta S2S_s(f, Proxy)$ and $\phi_{s2s_{cor.}(f,Proxy)}$ is the site-to-site variability of $\delta S2S_{s,cor.}(f, Proxy)$. $\delta S2S_{s,cor.}(f, Proxy)$ with the four site proxies are shown for $f = 0.529$, $1.062$ and $9.903\,\text{Hz}$ in Fig. A3 in the Appendix. If the site amplification model were able to perfectly predict and capture the full range of the site amplification at a specific site, $\phi_{s2s_{cor.}(f,Proxy)}$ would be reduced to zero. However, such an ideal case is not realistic and conventional site amplification models can only aim to reduce $\phi_{s2s_{cor.}(f,Proxy)}$ as much as possible. Hence, the greater the reduction in variability, meaning a lower $\phi_{s2s_{cor.}(f,Proxy)}$ the better the ability of the proxy to capture the site amplification. Still, it is important to keep in mind that this measure and the correlation between $\delta S2S_s(f)$ and the site proxies are purely statistical and do not give any insight into what is causing the amplification and its variability.

As described in Section 4, the site-amplification models and corresponding reduction in $\phi_{s2s}$ were derived and calculated 10 times following the 10-fold cross-validation technique. This means we are dealing with two sources of variability; the site-to-site variability $\phi_{s2s}$ and the variability related to running the regression on different subsets of the data in the 10-fold cross-correlation, here called $\epsilon_{cc}$. $\phi_{s2s}$ is a combination of the natural, random and irreducible (aleatory) variability of site response, and the epistemic uncertainty related to the site proxies not being able to fully capture the site properties controlling the site amplification. $\epsilon_{cc}$ is fully epistemic as it is related to the difference between the site-amplification models derived using different datasets. Figure 9 shows the mean $\phi_{s2s}$ for all stations (black lines) and $\phi_{s2s_{cor.}}$ (Proxy) for each proxy (coloured lines) from the 10 cross-validation iterations derived on the training sets (Fig. 9a) and the validation sets (Fig. 9b). The $\phi_{s2s}$ for each cross-validation iteration is shown in Fig. A4 in the Appendix. In Figure 9, the shaded areas around the means are the variance $\epsilon_{cc}$ related to the cross-validation process. $\epsilon_{cc}$ is, as expected, higher for the validation set (Fig. 9b), which for each

260 iteration corresponds to only 10 % of the dataset, than for the training set (90 % of the data, Fig. 9a). Nonetheless, the general pattern is similar for both sets, where the highest reduction in site-to-site variability is caused by the site-amplification model based on geological era and slope. Inferred $V_{S30}$, slope and geomorphological sediment thickness show similar reductions in variability for both the training and validation sets, with around a 1 % difference for the training set. For the validation set the reduction caused by Inferred $V_{S30}$, slope and geomorphological sediment thickness are within the same standard deviation $\epsilon_{cc}$

and are not clearly distinguishable (Fig. 9b). For both the training and validation set, none of the site-amplification models are distinguishable for frequencies above 3 Hz, which might be caused by the low resolution (30 arc seconds and 1:1500,000) of all the proxies considered in this study. This poor correlation between site amplification and topography- and geology-based proxies at high frequencies ($f > 3.0\,\mathrm{Hz}$) has also been observed by other studies Stewart et al. (2017); Zhu et al. (2022), also when higher resolution indirect proxies were used (Bergamo et al., 2022), indicating that inferred site proxies mainly captures

the average and deep properties of the subsurface and not the finer local nuances of a site.

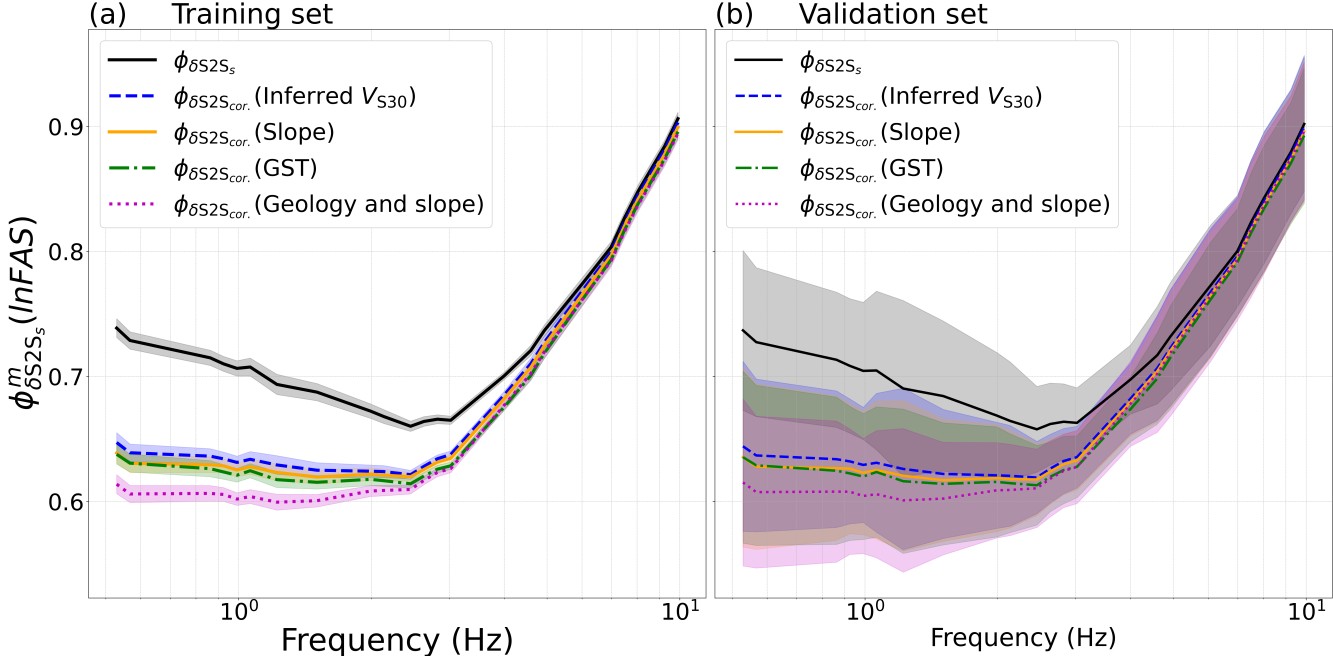

**Figure 9.** The site-to-site variability $\phi_{s2s}$ for all selected stations (solid black line) and the corrected site-to-site variability after subtracting the predicted site amplification using inferred $V_{S30}$ ($\phi_{s2s_{cor.}}(V_{S30})$), dashed blue lines), slope ($\phi_{s2s_{cor.}}$ (Slope), solid orange lines), geomorphological sediment thickness (GST) ($\phi_{s2s_{cor.}}$ (GST), dash-dotted green lines) and geology and slope ($\phi_{s2s_{cor.}}$ (Geology and slope), dotted magenta lines) from the empirical site amplification using (a) the 10-1 part training-set and (b) the 1 part validation-set.

The results shown in Fig. 9, indicate that the site-amplification model based on geological era and slope combined is better at capturing site amplification relative to the other proxies used in this study. These results are consistent with the findings of Weatherill et al. (2020), who also derived site-amplification models based on several inferred proxies for Japan. However,

when applying the same model to Europe, Weatherill et al. (2023) found that the reduction from geological era and slope was
not significantly lower than for inferred $V_{S30}$ or slope alone. They speculate that this could be an artefact of the mixed-effects
regression used to derive the model, where geological era and slope were included as a random effect, meaning the coefficient
for slope change with each geological era, which is different for the multiple linear regression applied in this study where the
slope coefficient stays the same for each geological era. In addition, our study uses $\delta S2S_s(f)$ from FAS, which is likely to
affect the site-to-site variability.

Nevertheless, the case that geological era and slope combined give the highest reduction to the site-to-site variability shows
the importance of including geology in site amplification modelling. Furthermore, the new proxy geomorphological sedimen-
tary thickness shows a similar or even slightly higher reduction in site-to-site variability as the traditional proxies inferred $V_{S30}$
and slope, indicating the potential of geomorphological sedimentary thickness as an alternative site proxy for seismic hazard
assessments for large areas or areas without measured site parameters.

## 6 Proxy-based site amplification predictions and maps

To further evaluate the ability of the site proxies to predict site amplification we compare the predicted site amplification with
empirical site amplification for the entire frequency range ($f = 0.460 - 9.903\,\mathrm{Hz}$). The object of this study is to test regionally
or globally available site proxies as predictors for regional site amplification over a large area. It is therefore not necessarily
meaningful to compare our site-amplification predictions to empirical site amplification at a single site. Instead, we compare
the predicted amplification to empirical amplification grouped according to the Eurocode 8 classes (EC8 CEN, 2004) provided
in the ESM database. Based on the station distribution with $V_{S30}$ (Fig. 3) and the number of stations per class (Fig. 10a), we
only use the stiffer classes, A, B and C, for the comparison. Selecting corresponding ranges of site properties for predicting the
site amplification for each EC8 class using other proxies than inferred $V_{S30}$, requires some attention. The correlation between
measured $V_{S30}$ and slope in Fig. 10b shows that slope generally increases with increasing $V_{S30}$. Although the relation has a
high variability, especially for high $V_{S30}$, we select high slope (0.1 – 0.3 m/m) for A, 0.05 – 0.1 m/m for B and 0.01 – 0.03 m/m
for C. Because the relation between geomorphological sediment thickness and measured $V_{S30}$ has a high variability over the
entire range (Fig. 10c) we use the newly proposed Eurocode 8 draft (Paolucci et al., 2021) with very shallow geomorphological
sediment thickness at ($< 5\,\mathrm{m}$) for A, shallow (5 - 30 m) and intermediate (30 - 50 m) for B and intermediate for C. When
selecting corresponding geological eras, we use Holocene and Pleistocene for A (Fig. 10b, yellow and pink scatter points), the
remaining geological eras (leaving out Cenozoic) Cretaceous, Jurassic-Triassic, Precambrian and Paleozoic for B (Fig. 10c,
brown, green, blue and purple scatter points), and Cretaceous and Jurassic-Triassic for A (Fig. 10c, brown and green scatter
points). The selected ranges of site properties are given in Table 1.

**Table 1.** The ranges of site properties selected to correspond with the Eurocode 8 classes; A, B and C.

| Proxy | A | | B | | C | |
|---|---|---|---|---|---|---|
| | Range | Number of stations | Range | Number of stations | Range | Number of stations |
| Measured $V_{S30}$ | > 800 m/s | 68 | 360 – 800 m/s | 184 | 180 – 360 m/s | 9 |
| Inferred $V_{S30}$ | > 800 m/s | 15 | 360 – 800m/s | 138 | 180 – 360 m/s | 65 |
| Slope | 0.1 – 0.3 m/m | 28 | 0.05 – 0.1 m/m | 69 | 0.01 – 0.03 m/m | 148 |
| GST | 0 – 5 m | 51 | 5-30 | 142 | 30 – 50 m | 311 |
| | | | 30 - 50 m | 7 | | |
| Geological era | Cretaceous | 9 | Cretaceous, | 22 | Holocene | 18 |
| | | | Jurassic-Triassic | 6 | | |
| | | | Precambrian | 1 | Pleistocene | 10 |
| | Jurassic-Triassic | 5 | Paleozoic | 1 | | |

Figure 11 shows the mean of the empirical site amplification (solid black line) with the standard deviation (shaded black area) of the sites in each of the EC8 classes A, B and C. The variability of the empirical site amplification even within each class is very high, and the predicted site amplifications using any of the proxies are within the standard deviation of the three classes. For A, slope and geomorphological sediment thickness are continuously in the upper range of the empirical site amplification standard deviation, while the predictions based on inferred $V_{S30}$ and geological era with slope are slightly under-predicting relative to the mean empirical site amplification for low frequencies and over-predicting for higher frequencies. For class B, inferred $V_{S30}$. and slope are in the lower range of the empirical site amplification standard deviation but generally follow the shape of the mean empirical site amplification, while the prediction based on shallow (5 - 30 m) and especially intermediate (30 - 50 m) geomorphological sediment thickness are over-predicting with respect to the mean empirical amplification, The site amplification models based on geological era and slope are under-predicting relative to the mean empirical site amplification, especially for low frequencies, but gets closer to the mean empirical amplification at higher frequencies. For the softer soil class C, the different proxies have similar predictions and are close to the mean of the empirical amplification. At low frequencies, in particular the model predictions based on geomorphological sediment thickness and inferred $V_{S30}$, are close to the mean empirical amplification. However, the differences between the predictions and empirical amplification could also be due to a poor correspondence between the selected ranges of inferred proxy values and the measured $V_{S30}$, particularly for geological era.

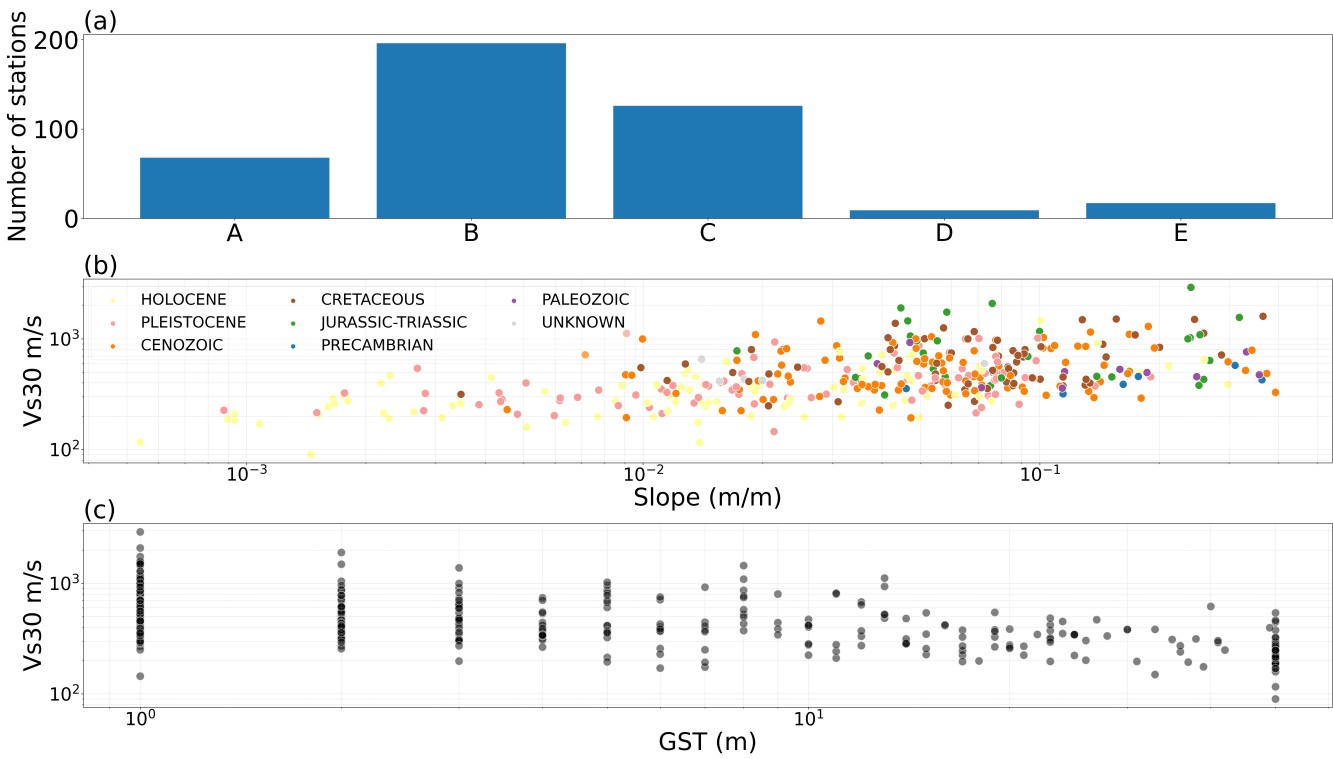

**Figure 10.** (a) Number of stations per Eurocode 8 class, (b) The correlation between measured $V_{S30}$ and slope coloured by geological era and (c) geomorphological sediment thickness (GST).

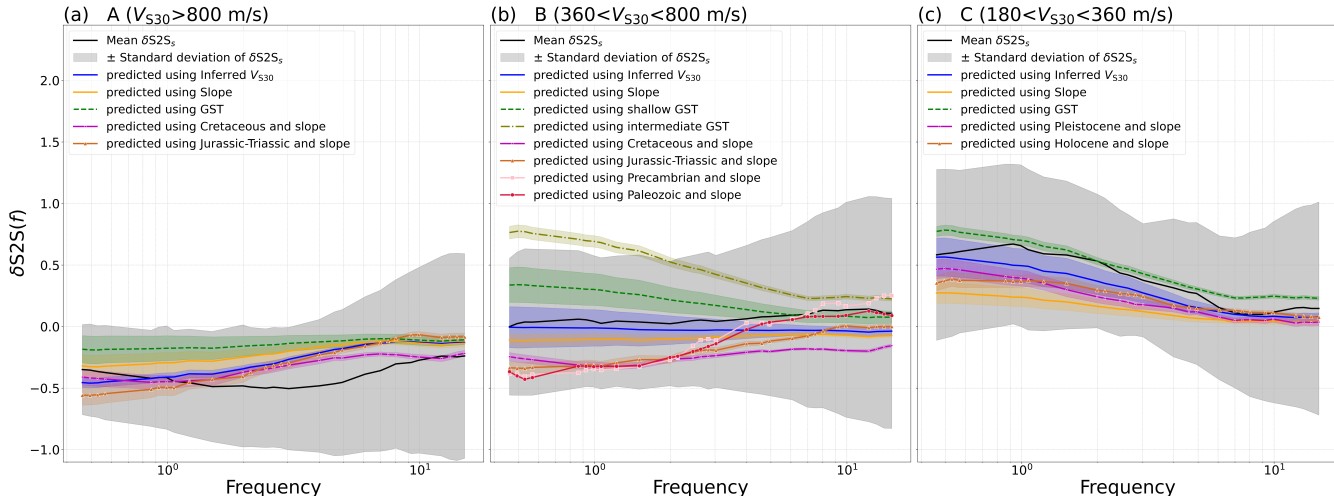

**Figure 11.** Empirical mean (solid black line) site amplification with standard deviation (shaded black area) compared to predicted site amplification for the Eurocode classes (a) A, (b) B and (c) C, using inferred $V_{S30}$ (dashed blue lines), slope (solid orange lines), geomorphological sediment thickness (dash-dotted green and olive lines) and geological era and slope (dotted magenta lines).

Finally, we use the proxy-based site-amplification models to predict the site amplification for the entire Europe, as shown in Fig. 12 for $f = 1.062\,\mathrm{Hz}$. The amplification maps at $f = 0.529\,\mathrm{Hz}$ and $f = 9.903\,\mathrm{Hz}$ are shown in Figures A5 and A6 in the Appendix. The site-amplification maps show the amplification (red) and deamplification (blue) relative to the median site amplification (white) predicted by the associated GMM defined in Eq. (1 - 4). This is different from conventional amplification maps used in PSHA, where the site amplification is relative to a rock reference. In this study, we keep the amplification relative to the median to avoid biasing the result with poorly constrained rock properties. Furthermore, the $\delta S2S_s$ used to develop the models are obtained from strong-motion stations mainly located in Southern Europe and the Mediterranean region (Fig. 1), some regional bias is therefore likely to be present in the amplification predictions. The maps all show similar features, for example, high amplification in the soft sediment basins of the Po-Plains in northern Italy, the Danube plains of Eastern Romania and the Great Hungarian plains, and strong deamplification in the Alps, the Carpathian Mountains and Western Norway. However, clear differences are also evident in the different site-amplification maps, for example around the Rhine valley, in the Baltic and eastern Türkiye. These differences show that the proxies are capturing different aspects of the site effects, but also that there is a need for characterising the epistemic uncertainty related to the site amplification predictions.

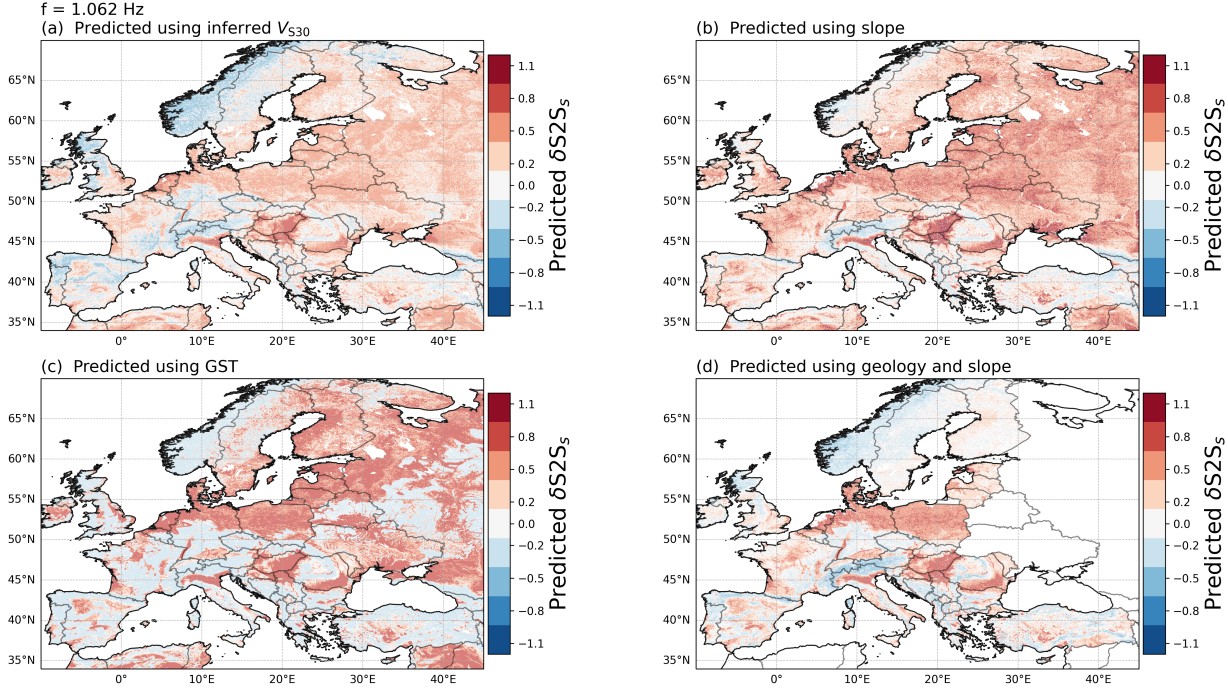

**Figure 12.** Predicted site amplification at $f = 1.062\,\mathrm{Hz}$ for Europe using (a) inferred $V_{S30}$ (b) slope, (c) geomorphological sedimentary thickness, and (d) geological era and slope. The amplification maps are relative to the median prediction of associated GMM given in Eq. (1 - 4)

### 6.1 Justifications of the differences in the site amplification predictions: A focus on eastern Türkiye

To investigate the differences in the predictions further, and evaluate how the models perform with new data, we zoom in on Eastern Türkiye where the recent Kahramanmaraş Earthquake Sequence of February 2023 occurred (Melgar et al., 2023; Petersen et al., 2023). Figures 13 and 14 show the site proxies and predicted site amplification at $f = 1.062\,\mathrm{Hz}$ in Eastern Türkiye and Syria. The main difference between the site proxies and their corresponding site amplification is in the area south of the southeastern Taurus Mountains, around the cities Gaziantep and Aleppo, and by the border between Türkiye and Syria. In the map based on inferred $V_{S30}$ (Fig. 13a) and slope (Fig. 13b), low values of $V_{S30}$ and slope are present on both sides of the border and in the corresponding amplification maps (Fig. 14a and b), medium to high ($0.2 < \delta S2S_s < 0.6$) amplification is predicted around the border with gradual lower amplification towards the Taurus Mountains. The map of geomorphological sedimentary thickness (Fig. 13c) differs markedly from that of inferred $V_{S30}$ and slope north of the Türkiye-Syria border with mainly shallow ($< 10\,\mathrm{m}$) sediments and low amplification ($-0.6 < \delta S2S_s < 0$) is predicted for that area (Fig. 14c). South of the Türkiye-Syria border, however, the geomorphological sedimentary thickness map shows deep ($> 30\,\mathrm{m}$) sediments and predicts high amplification ($\delta S2S_s > 0.6$). This sharp difference at the Türkiye-Syria border in the geomorphological sedimentary thickness is likely due to the fact that Pelletier et al. (2016), in the process of differentiating between upland and lowland, used

separate geologic maps for Europe and the Arabian Peninsula. The site amplification map based on geological era and slope combined (Fig. 14d) predicts less high amplification than inferred $V_{S30}$ and slope and mainly in concentrated areas north of the Türkiye-Syria border (Fig. 14a and b). However, because the geological era maps were created for the European Seismic Hazard and Risk model, the maps based on geological era and slope end at the border of Türkiye and do not include Syria (Fig.

13d). Despite these limitations, we chose to focus on this area because the recent events caused large damages on both sides of the borders, showing the urgent need for Seismic Hazard and Risk models that cross both national and regional borders.

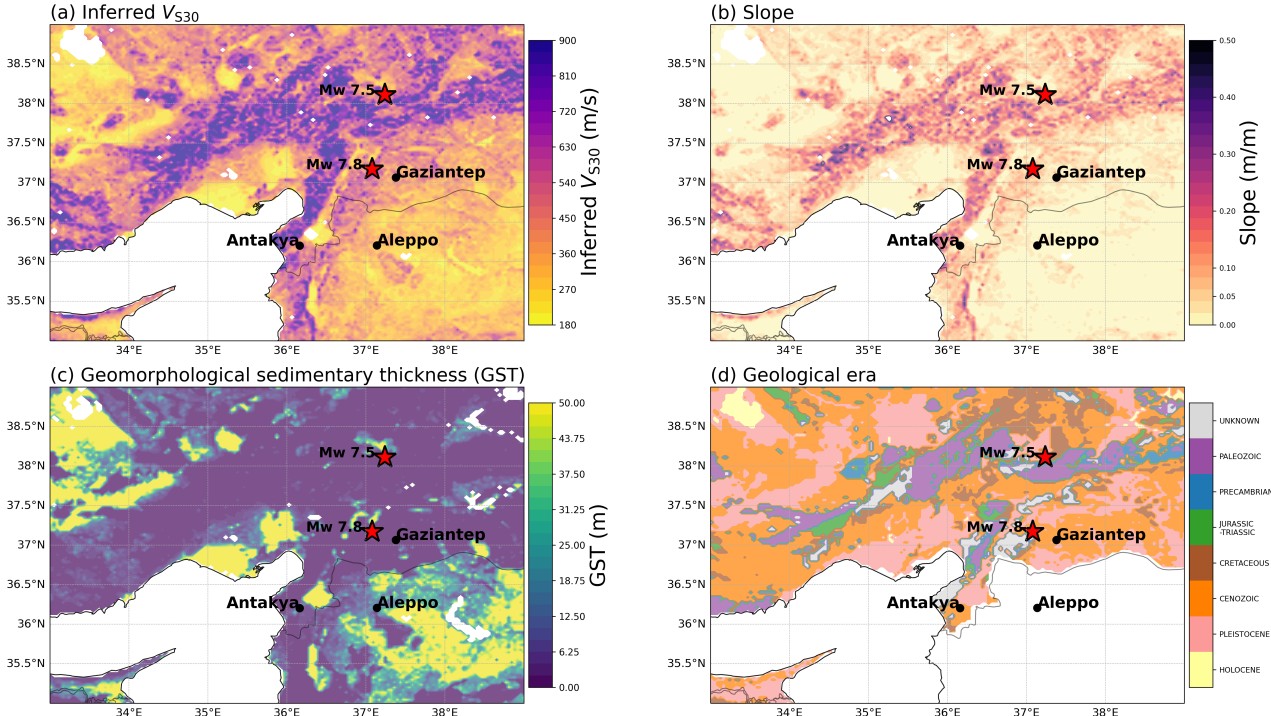

**Figure 13.** Map of Eastern Türkiye and Syria coloured by the site-proxies (a) inferred $V_{S30}$ (b) slope, (c) geomorphological sedimentary thickness, and (d) geological era. The epicentres for the two largest earthquakes of the Kahramanmaraş Sequence of February 2023 are indicated as red stars on the map.

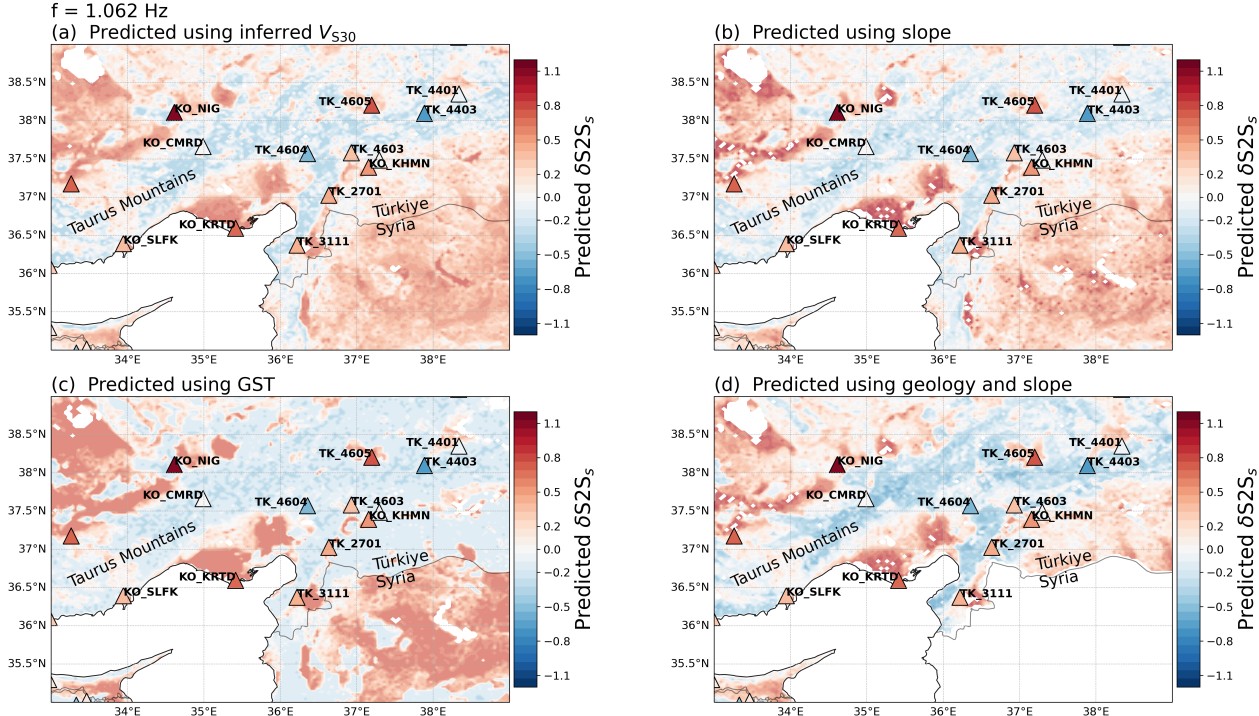

**Figure 14.** Predicted site amplification in Eastern Türkiye and Syria at $f = 1.062\,\text{Hz}$ using (a) inferred $V_{S30}$ (b) slope, (c) geomorphological sedimentary thickness, and (d) geological era and slope. The triangles represent the strong motion stations coloured by empirical site amplification $\delta S2S_s$ at $f = 1.062\,\text{Hz}$. The stations with measured $V_{S30}$ values are indicated on the map. The amplification maps are relative to the median prediction of associated GMM given in Eq. (1 - 4)

We use ground motions from eastern Türkiye recorded in February and March 2023 to evaluate the performance of the models. The new dataset was retrieved from the ESM database (Luzi et al., 2020), and contains events recorded by 290 stations. The distribution and map of the events and stations are shown in Fig. 15. The predicted ground motion at each site is obtained using the four proxy-based site amplification models ($Y_{s,Proxy}$) in combination with the median GMM prediction ($\mu_{median}$):

$$\ln(\mu_{s,Proxy}) = \ln(\exp(\mu_{median} + Y_{s,Proxy}) \tag{9}$$

$$res = \ln(FAS) - \ln(\mu_{s,Proxy}) = \delta B_e + \delta W_s \tag{10}$$

As the majority of the stations in the new dataset did not record a sufficient number of events ($> 3$) to derive the site-to-site residual $\delta S2S_s$, we only use event as a random effect and evaluate the predictability of the four models using the within-event

residual $\delta W_s$. Figure 16 shows the site corrected $\delta W_s$ obtained from the GMM predictions of the Kahramanmaraş Earthquake Sequence and the four different site amplification models at $f = 1.062\,\mathrm{Hz}$. The $\delta W_s$ for $f = 0.529\,\mathrm{Hz}$ and $f = 9.903\,\mathrm{Hz}$ are shown in Figures A7 and A8 in the Appendix. The variability of the site corrected $\delta W_s$ is high for all the proxies, partly because the record-to-record variability $\delta WS_{e,s}$ is also included in the $\delta W_s$, whereas the models are only predicting $\delta S2S_s$.

Nevertheless, the site-corrected $\delta W_s$ for all the proxies are consistently centred around zero with no visible strong trends, as shown by the binned mean (blue error bars). This shows that no significant biases are present in the models and that all four proxies are satisfactorily able to predict site amplification, even for new data. However, due to the limitations of the dataset, any further conclusions on the site effects caused by these events are beyond the scope of this study.

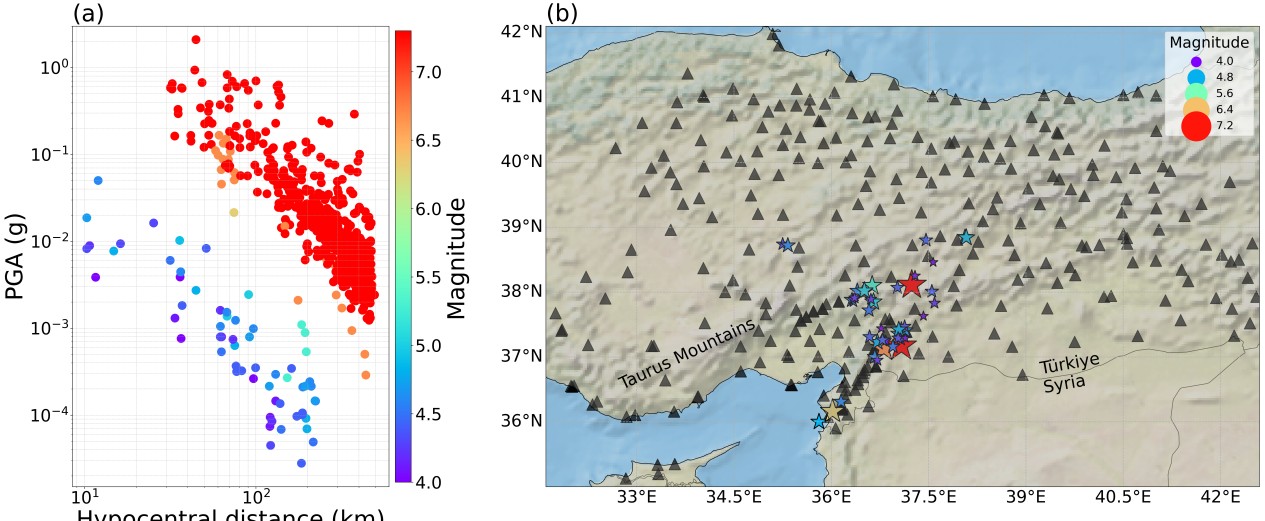

**Figure 15.** (a) Distribution and (b) map of the ground motions from eastern Türkiye recorded in February and March 2023.

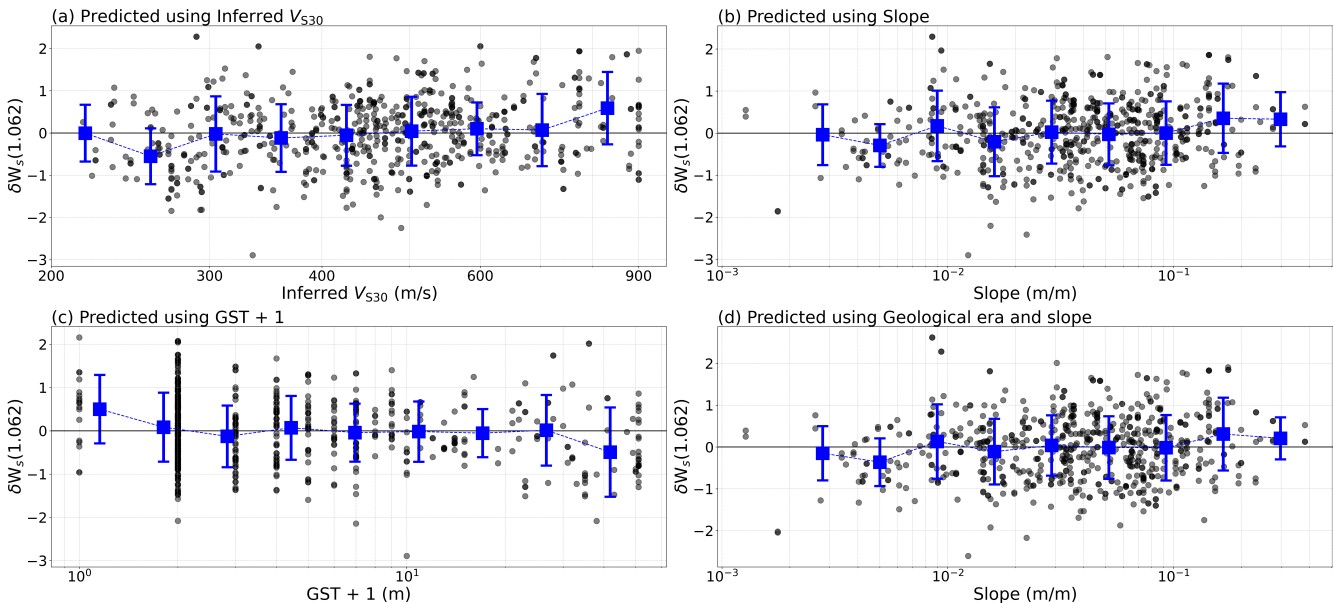

**Figure 16.** The within-event residuals $\delta W_s$ from predicted ground motions (in FAS) at $f = 1.062\,\mathrm{Hz}$ of the recent Kahramanmaras Earthquake Sequence of February 2023 using the site amplification models based on (a) inferred $V_{\mathrm{S30}}$ (b) slope, (c) geomorphological sedimentary thickness, and (d) geological era and slope.

Furthermore, because there are few stations with calculated $\delta S2S_s$ in this area, and particularly where the largest differences are located, it is difficult to make a solid argument for which proxy is more correct. For this purpose, one would either have to wait for more events to derive a valid $\delta S2S_s$, or perform a more detailed comparison with local values. However, as stated before, the aim of this study is not to re-create the exact site-specific amplification, but to evaluate the ability of inferred proxies to predict the site amplification for larger areas and capture, using several models, the epistemic uncertainty of such regional amplification prediction. Instead, we compare the empirical amplification from the few stations in the area with the predicted amplification in a similar way as in Fig. 11. We only use the 14 stations available in the original $\delta S2S_s$ dataset and with measured $V_{\mathrm{S30}}$ values, these stations are marked in Fig. 14 and are described in Table 2. Because the measured $V_{\mathrm{S30}}$ of the considered stations only range between 300 and 700 m/s, the stations are separated into two ranges for medium stiff soil sites ($V_{\mathrm{S30}}$ = 300-500 m/s, Fig. 17a) and stiffer sites ($V_{\mathrm{S30}}$ = 500-700m/s, Fig. 17b). The comparison of empirical amplification and predicted amplification for the East-Turkish stations in the two $V_{\mathrm{S30}}$ ranges shows that all the proxies are significantly under-predicting the site amplification for the medium stiff soil sites and over-predicting the stiffer sites. Figure 17 further shows that predicting site amplification using proxies cannot reproduce the full range of amplifications on a local level and that regionalized models might be necessary.

In addition to assessing whether new site proxies can provide additional valuable information for estimating site effects at regional or global scales, an important motivation for this and previous studies (Weatherill et al., 2020, 2023) was to discuss how to deal with epistemic uncertainty when characterising ground motion at sites for further probabilistic seismic hazard

and risk calculations. In order to properly account for the uncertainty, it is necessary to have a clear picture of where the uncertainty comes from. The results presented above have shown that using different site proxies to predict site amplification gives significantly different results, which further emphasises the importance of capturing the epistemic uncertainty associated with modelling site amplification when using inferred proxies. Furthermore, this epistemic uncertainty needs to be incorporated into the final risk calculation. To fully assess the impact of this epistemic uncertainty, risk and loss calculations should be performed using the different site amplification models, but this is beyond the scope of this work.

Table 2. Station name, location and site properties of the East Turkish stations with measured $V_{S30}$ shown in Figure 14.

| Station code | Latitude | Longitude | $V_{S30}$ (m/s) | $V_{S30,WA}$ (m/s) | Slope (m/m) | GST (m) | Geological era |
|---|---|---|---|---|---|---|---|
| TK-4401 | 38.34962 | 38.34019 | 481.0 | 502.0 | 0.069 | 3.0 | Cenozoic |
| TK-3102 | 36.21300 | 36.15900 | 469.0 | 340.0 | 0.009 | 27.0 | Cenozoic |
| TK-4403 | 38.09616 | 37.88732 | 655.0 | 425.0 | 0.014 | 1.0 | Unknown |
| TK-4605 | 38.20368 | 37.19771 | 315.0 | 313.0 | 0.003 | 38.0 | Cretaceous |
| TK-4603 | 37.57998 | 36.93061 | 465.0 | 523.0 | 0.049 | 3.0 | Pleistocene |
| TK-4604 | 37.57010 | 36.35737 | 613.0 | 625.0 | 0.064 | 2.0 | Cenozoic |
| TK-3101 | 36.21423 | 36.15973 | 469.0 | 340.0 | 0.009 | 27.0 | Cenozoic |
| TK-3103 | 36.11593 | 36.24722 | 344.0 | 512.0 | 0.044 | 4.0 | Cenozoic |
| TK-3105 | 36.80262 | 36.51119 | 619.0 | 602.0 | 0.078 | 1.0 | Pleistocene |
| TK-3111 | 36.37260 | 36.21973 | 338.0 | 619.0 | 0.051 | 4.0 | Pleistocene |
| TK-2701 | 37.02546 | 36.63593 | 421.0 | 457.0 | 0.020 | 16.0 | Unknown |
| TK-4607 | 37.48513 | 37.29775 | 672.0 | 501.0 | 0.059 | - | Cenozoic |
| TK-2702 | 37.18430 | 36.73280 | 599.0 | 469.0 | 0.073 | 3.0 | Unknown |
| TK-4601 | 37.53872 | 36.98187 | 345.0 | 349.0 | 0.017 | 22.0 | Pleistocene |

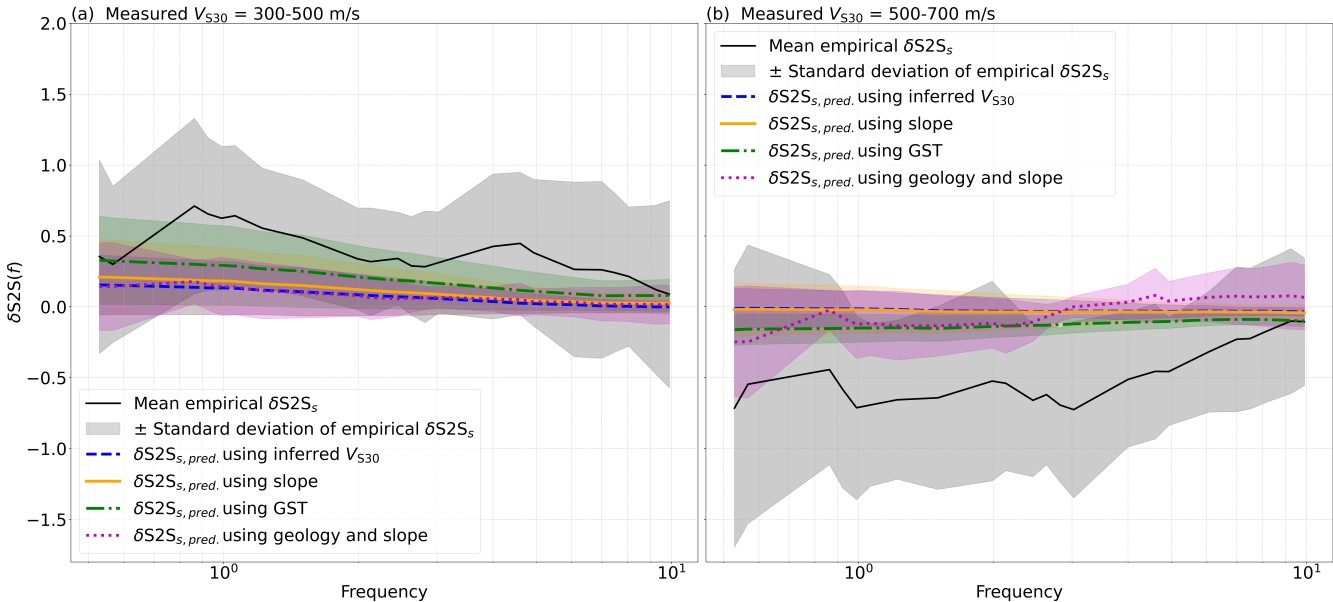

**Figure 17.** empirical site amplification (solid black line) compared to predicted site amplification using the stations in Eastern-Türkiye with measured $V_{S30}$-values, separated into medium stiff soil sites with (a) $V_{S30} = 300 - 500$ m/s and (b) stiff sites $V_{S30} = 500 - 700$ m/s, using inferred $V_{S30}$ (dashed blue lines), slope (solid orange lines), geomorphological sediment thickness (dashed green lines) and geological era and slope (dotted magenta lines) at the stations.

## 7  Conclusions

To test whether the geomorphological model for sediment thickness derived by Pelletier et al. (2016) can be used as an alternative site proxy, we have derived site-amplification models based on the geomorphological sediment thickness, as well as tra-

ditional site proxies like inferred $V_{S30}$, slope and geological era and slope combined. Although the predicted site-amplification maps based on the different proxies show similar trends, there are also notable differences, indicating that the proxies capture different aspects of the site effects. Using only one proxy, for example, inferred $V_{S30}$ which is often the standard procedure, is therefore not sufficient to fully capture the range of possible amplifications. The differences in the site-amplification predictions based on the different proxies thus contribute to the characterisation of the epistemic uncertainty. In a probabilistic

seismic hazard and risk context, this uncertainty needs to be included and properly accounted for. In this study, we calculate the site amplification for Europe and the Middle East but focus particularly on Eastern Türkiye and Syria. As a measure of how well the site proxies capture the empirical site amplification we used the reduction in site-to-site variability. The results show that the site-amplification predictions based on geological era and slope combined cause the highest reduction while the prediction based on geomorphological sediment thickness causes a similar, but slightly larger, reduction in site-to-site variability

than the traditional site proxies, inferred $V_{S30}$ and slope. This result shows the value of including geology and geomorphology in prediction models for site amplification. Furthermore, the geological map used in this study is only available for Europe,

while geomorphological sediment thickness is available globally and easily accessible. However, although the geomorphological sediment thickness has potential, further investigations and tests are needed before establishing it as an alternative to the much-used inferred $V_{S30}$ model from Wald and Allen (2007), in particular in areas where inferred $V_{S30}$ and slope are known to have a weak correlation with site amplification. Moreover, the correlation between the empirical site amplification and the site proxies all weakens above 3 Hz, which shows the need for models with higher resolution or including more local and shallow information. Our results therefore show the potential, of not only the geomorphological sediment thickness model but also of other models for soil and sediment thickness from geomorphology and similar fields outside seismology and earthquake engineering. The outputs of this study have been made available, see the Code and Data availability section below, however, it is important to state that the site amplification models and maps developed in this study can only be interpreted as referenced to the associated GMM median prediction, and should be considered as exploratory as they were developed for the purpose of testing the different site proxies and show the epistemic uncertainty related to using different proxies. Additionally, using inferred site proxies should only be done for regional seismic hazard studies of larger areas or when more detailed site parameters are missing.

*Code and data availability.* The empirical site amplification $\delta S2S_s$ and the coefficients for the proxy-based site amplification prediction models derived in this study, as well as a short code for computing and mapping the proxies and site-amplification predictions, are available on Zenodo: https://zenodo.org/record/8140143. The $V_{S30}$ dataset from Wald and Allen (2007) is available from the U.S. Geological Survey (USGS) and was downloaded from https://earthquake.usgs.gov/static/lfs/data/$V_{S30}$/$V_{S30}$.zip (last accessed 19.01.2023). The slope and geological map of Europe are available from the EFEHR seismic risk web services (http://risk.efehr.org/site-model/) and can be downloaded from https://maps.eu-risk.eucentre.it/map/european-site-response-model-datasets/download (last accessed 23.09.2022), and https://nextcloud.gfz-potsdam.de/s/93ZR4ky8D4mDXb9 (last accessed 02.05.2022), respectively. The geomorphological sediment thickness from Pelletier et al. (2016) can be downloaded from https://daac.ornl.gov/cgi-bin/dsviewer.pl?ds_id=1304 (last accessed 26.01.2021).

**Appendix: List of Tables**

**Appendix: List of Figures**

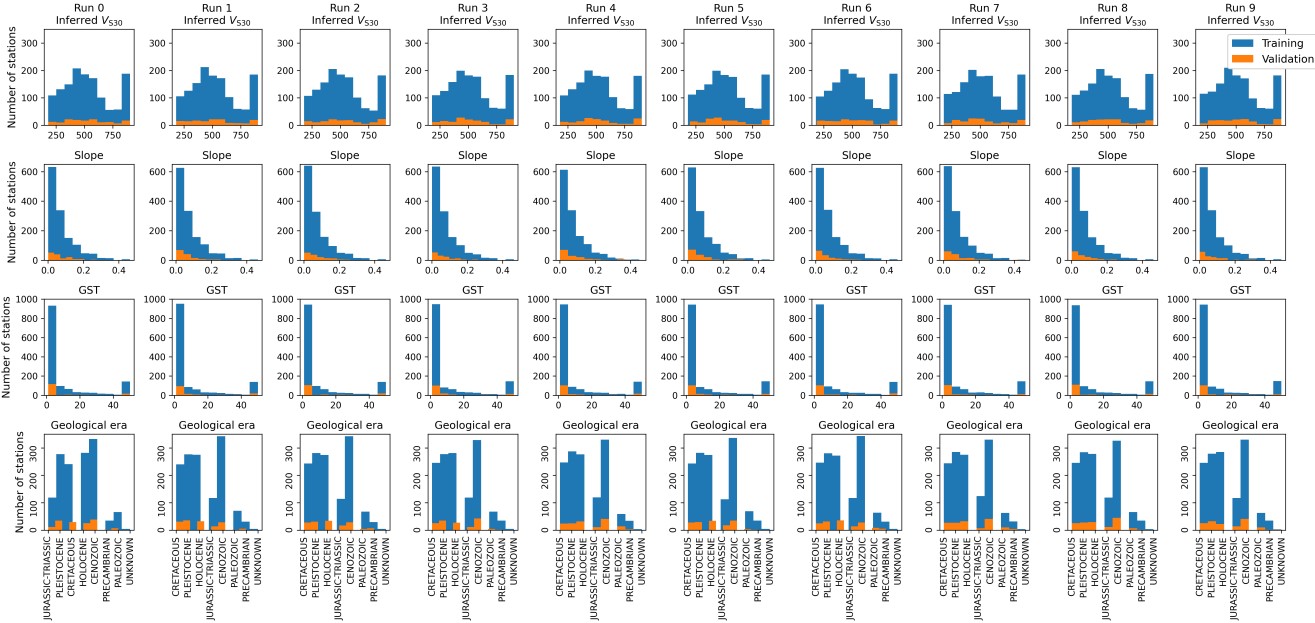

**Figure A1.** The distribution of inferred $V_{\text{S30}}$ (top row), slope (second row), geomorphological sediment thickness (GST, third row) and geological era and slope (bottom row) for each 10 fold cross validation iteration using the 10-1 part training-set (blue) and the 1 part validation-set (orange).

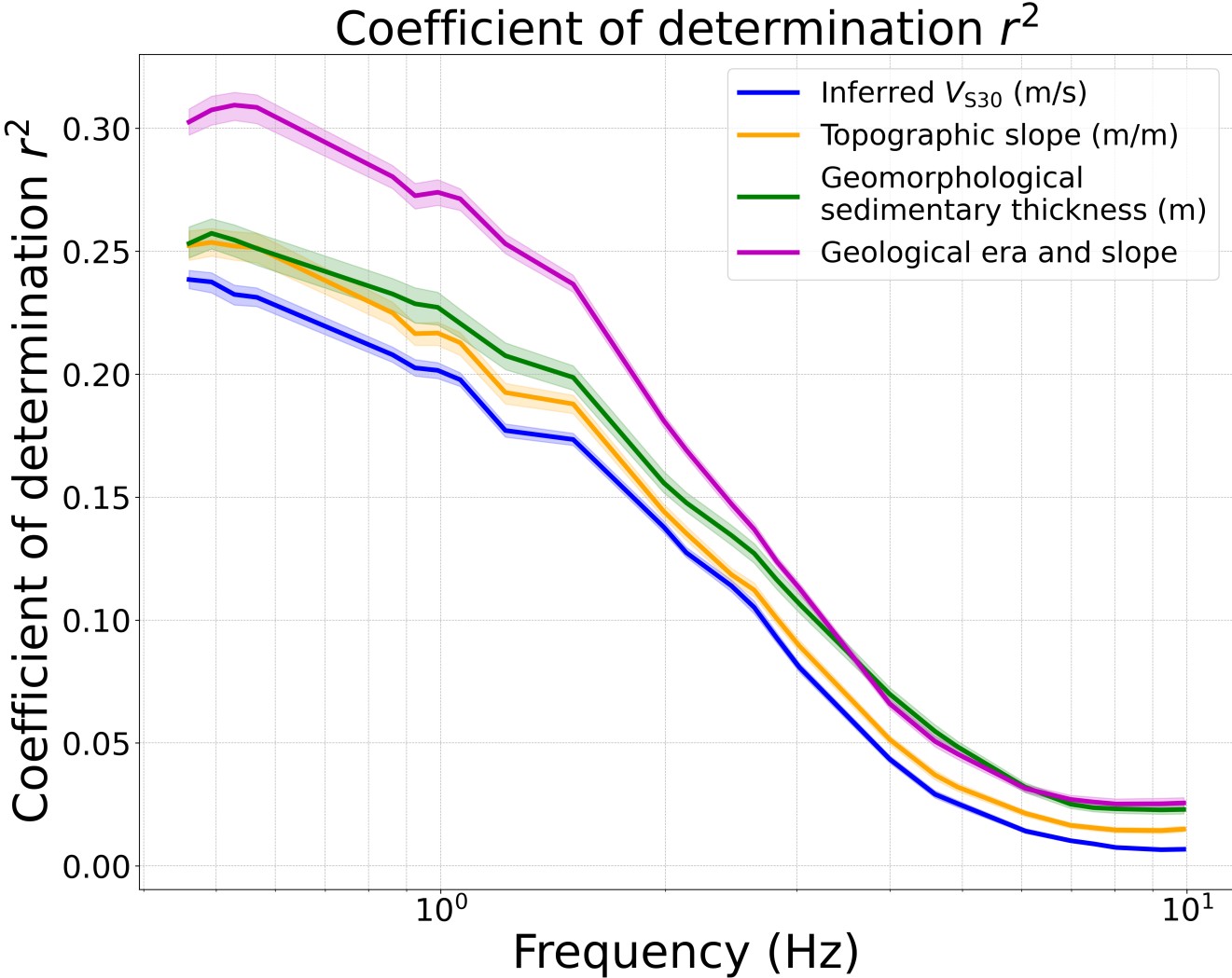

**Figure A2.** Coefficient of determination $r^2$ for the linear regressions between $\delta S2S_s(f)$ and the site proxies inferred $V_{S30}$ (blue line) slope (yellow) geomorphological sedimentary thickness (green) and geological era and slope (magenta) for the entire frequency range, as shown in Fig. 7 and 8 for $f = 0.529$ and $1.062\,\mathrm{Hz}$.

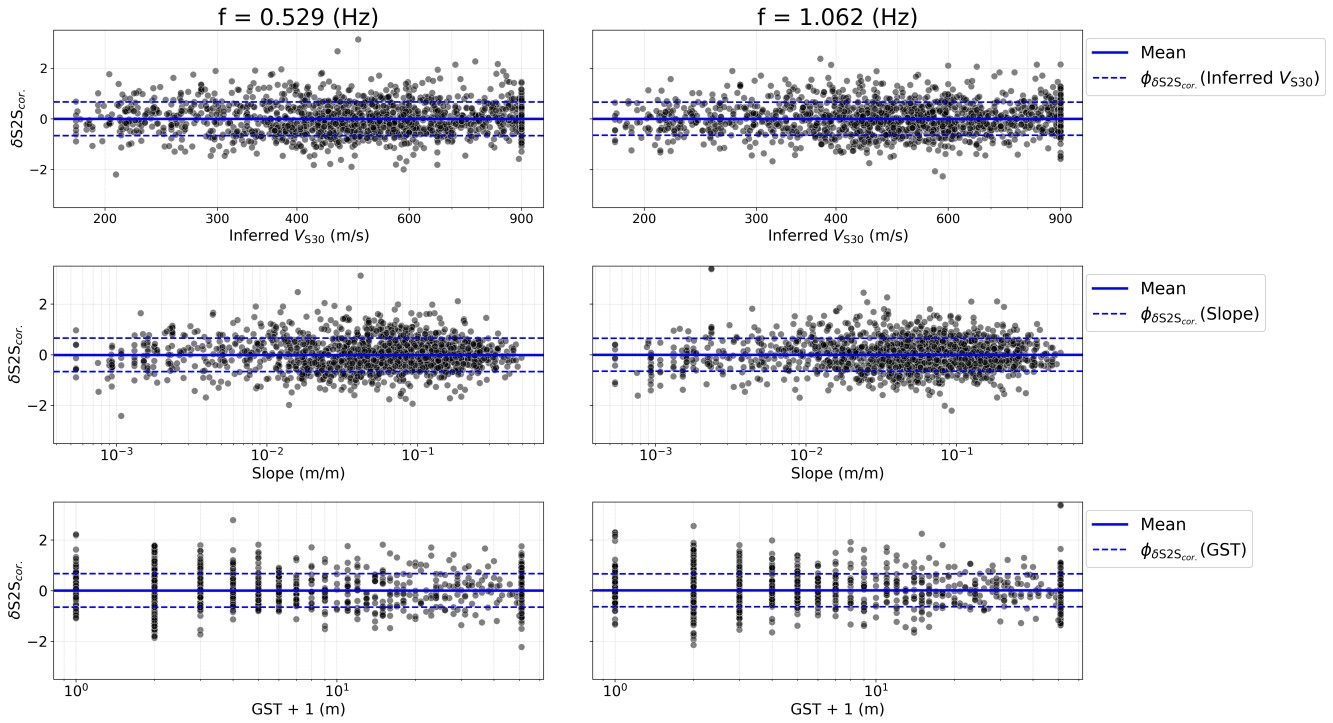

**Figure A3.** The site corrected $\delta S2S_{s,cor.}(f, Proxy)$ for $f = 0.529\,\text{Hz}$ (left), $f = 1.062\,\text{Hz}$ and $f = 9.903\,\text{Hz}$ for with inferred $V_{\text{S30}}$ (top row) slope (second top row), geomorphological sedimentary thickness (second bottom row) and geological era and slope (bottom row).

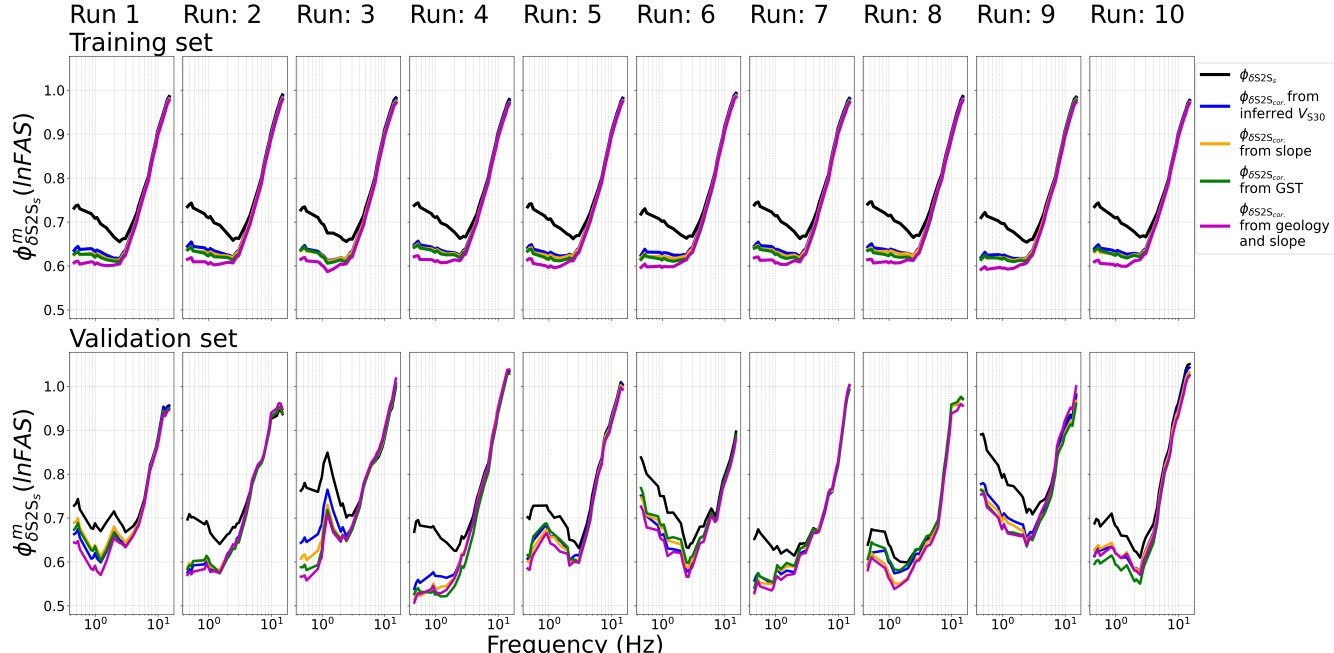

**Figure A4.** The site-to-site variability $\phi_{s2s}$ for all selected stations (black line) and the corrected site-to-site variability after subtracting the predicted site amplification using inferred $V_{S30}$ ($\phi_{s2s_{cor.}}(V_{S30})$, blue lines), slope ($\phi_{s2s_{cor.}}(\text{Slope})$, orange lines), geomorphological sediment thickness (GST) ($\phi_{s2s_{cor.}}(\text{GST})$, green lines) and geological era and slope ($\phi_{s2s_{cor.}}(\text{Geology and slope})$, magenta lines) from the empirical site amplification for each 10-fold cross-validation iteration using the $10-1$ part training set (top row) and the 1 part validation set (bottom row).

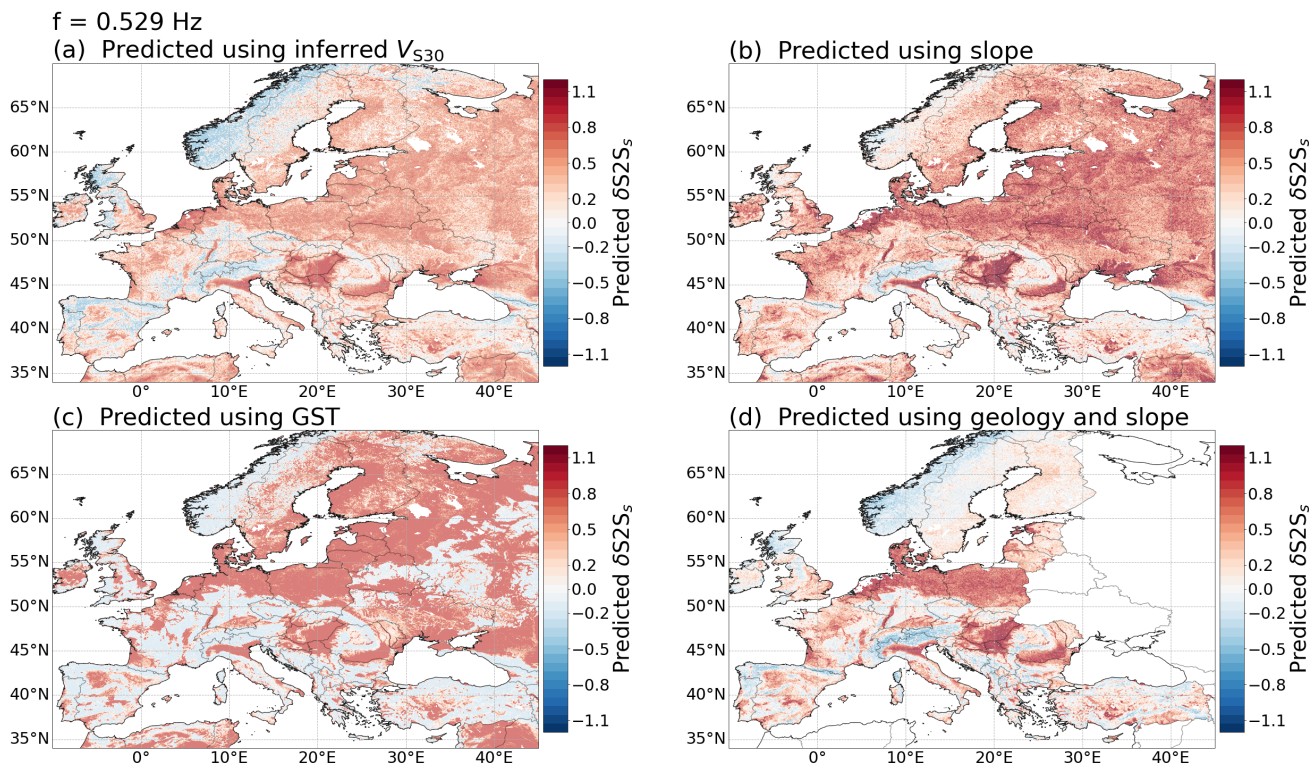

**Figure A5.** Predicted site amplification at $f = 0.529\,\mathrm{Hz}$ for Europe using (a) inferred $V_{S30}$ (b) slope, (c) geomorphological sedimentary thickness, and (d) geological era and slope. The amplification maps are relative to the median prediction of the associated GMM given in Eq. (1 - 4)

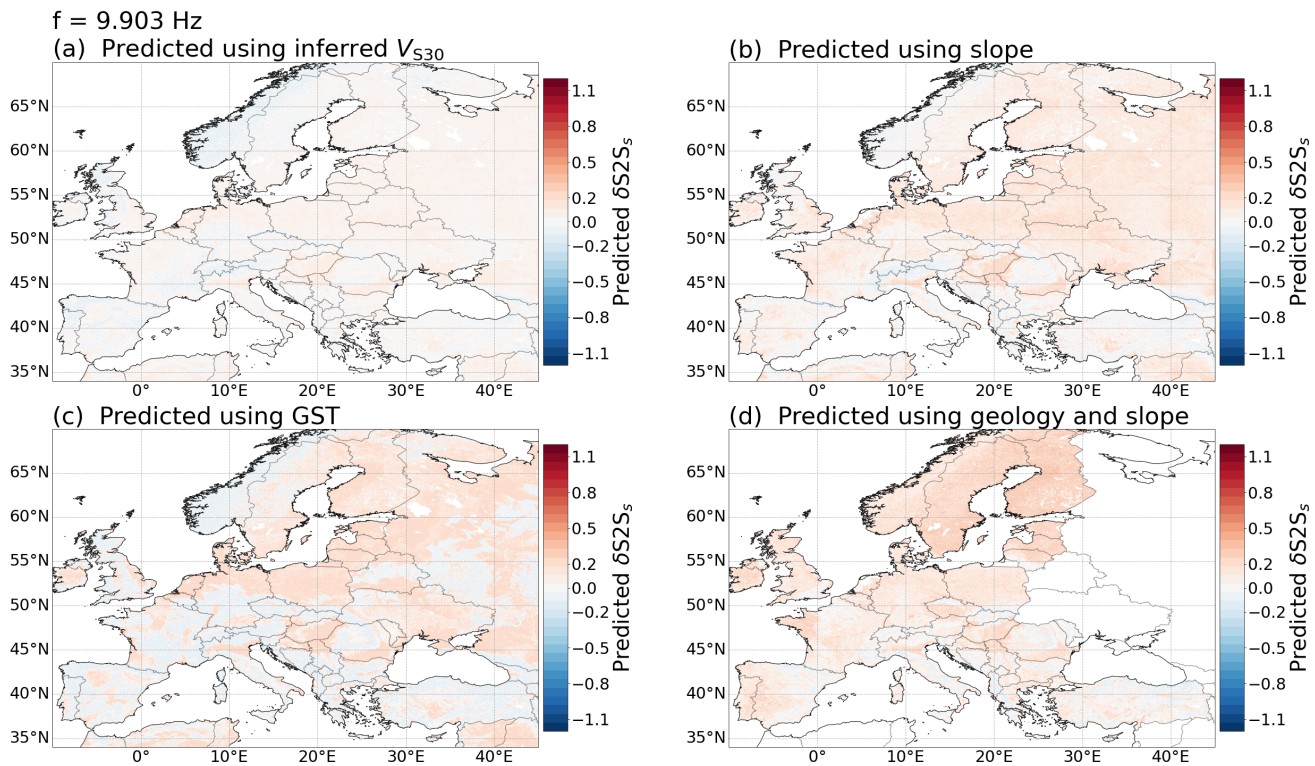

**Figure A6.** Predicted site amplification at $f = 9.903\,\text{Hz}$ for Europe using (a) inferred $V_{\text{S30}}$ (b) slope, (c) geomorphological sedimentary thickness, and (d) geological era and slope. The amplification maps are relative to the median prediction of the associated GMM given in Eq. (1 - 4)

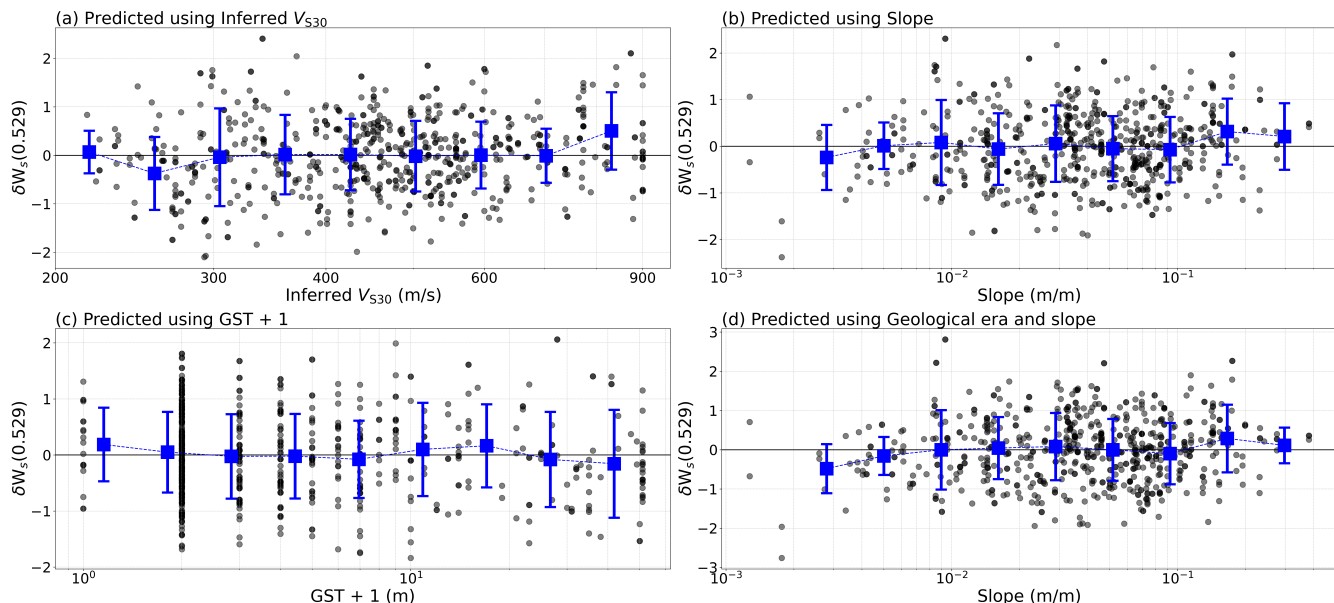

**Figure A7.** The within-event residuals $\delta W_s$ from the predicted ground motions (in FAS) at $f = 0.529\,\text{Hz}$ of the recent Kahramanmaras Earthquake Sequence of February 2023 using the site amplification models based on (a) inferred $V_{S30}$ (b) slope, (c) geomorphological sedimentary thickness, and (d) geological era and slope.

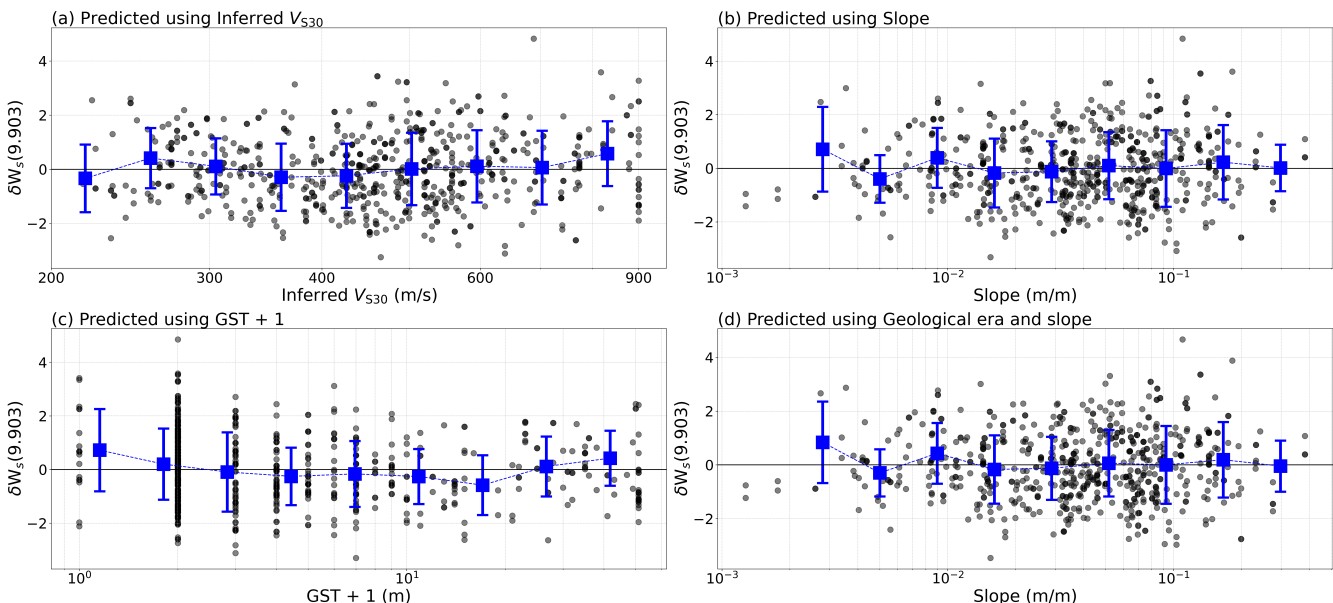

**Figure A8.** The within-event residuals $\delta W_s$ from the predicted ground motions (in FAS) at $f = 9.903\,\mathrm{Hz}$ of the recent Kahramanmaras Earthquake Sequence of February 2023 using the site amplification models based on (a) inferred $V_{S30}$ (b) slope, (c) geomorphological sedimentary thickness, and (d) geological era and slope.

*Author contributions.* TEXT

Karina Loviknes: Conceptualization, methodology, data preparation, investigation, visualization, writing. Fabrice Cotton: Conceptualization, methodology, supervision, review, and editing. Graeme Weatherill: Conceptualization, data preparation, review, and editing

*Competing interests.* TEXT

Fabrice Cotton is a member of the editorial committee for this special issue of Natural Hazards and Earth System Sciences and therefore cannot be chosen as the editor in charge of the paper.

*Disclaimer.* TEXT

*Acknowledgements.* The authors are grateful to Jean Braun for the valuable discussion and explanation on regolith. This research is funded by the European Commission, ITN-Marie Sklodowska-Curie New Challenges for Urban Engineering Seismology URBASIS-EU project, under Grant Agreement 813137 and supported within the funding program "Open Access publication costs" Deutsche Forschungsgemeinschaft (DFG, German Research Foundation) - Project Number 491075472.

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
