# Peer review of "Exploring inferred geomorphological sediment thickness as a new site proxy to predict ground-shaking amplification at regional scale. Application to Europe and Eastern Türkiye"

_EGUsphere, 2023_

## Referee Comment (RC1)

**Introducing inferred geomorphological sediment thickness as a new site proxy to predict ground-shaking amplification at regional scale. Application to Europe and Eastern Turkey**

**Karina Loviknes, Fabrice Cotton, Graeme Weatherill**
* * *
The authors have tested the geomorphological sediment thickness (GST) and classical local soil-conditions proxies such as Vs30 in response to this question: Is GST a relevant global site proxy for PSHA analysis.  To do this, they used site-to-site residuals issued from Kotha FAS-Model.

The selected study areas and proposed methodology are of scientific and engineering interest, aligned with the scope of the natural hazards and earth system sciences.

However, the claim that the developed model is based on site-to-site residuals using linear regression is questioned, the organization of the different sections can be improved, and the manuscript lacks adequate explanation in a few areas for a reviewer to fairly assess the technical quality of the study:

**1. General remarks**

In my opinion, the use of delta_S2S residuals (which is considered here as an epistemic uncertainty) is strongly linked to the Kotha model. To validate the method, several GMMs need to be considered and it needs to be proven that the delta_S2S values do not depend too much on the GMM used.

For me, the simplest way is to deduce the amplification from the FAS ratio and to find a correlation between this ratio and the site proxies. e.g. if this ratio = 2 its interpretation is simple, but with a delta_S2S=-1.5 how can we interpret it especially this study targets ESHM20 by proposing a regional proxy (GST).

With this study we remark that Delta_S2S=f(Proxy) is an heteroscedastic model (Khota model). This Heteroscedasticity lead to biased estimates of the standard errors of the regression coefficients. This can make it difficult to determine the significance of the coefficients and can lead to incorrect conclusions about the relationship between the proxy in the model.

Additionally, heteroscedasticity can also affect the efficiency of the parameter estimates, leading to less precise estimates than would be obtained with a homoscedastic model. I know you are looking for this heteroscedasticity, since the GMM (Kotha) model does not contain any proxy representing site effect. However, I'm wondering if, can you add a site proxy (e.g. GST) to the GMM model to ensure that the delta_S2S follows a homoscedastic model (goal is to validate GMM model).

**Why** did you use linear regression to have Site amplification factor ?. Site behavior is so complicated that a simple linear model is insufficient to consider the underlying physics. Or use small strain conditions to remove nonlinear effect. In addition, delta_S2S. In this sense, it must be shown that the model developed does not suffer from underfitting.

And I wonder how to obtain the amplification value with delta_S2S. Finally, if this is a sensitivity study, it should be mentioned in the title. Like : **Using site-to-site residuals to testing the Relevance of geomorphological sediment thickness as a regional site proxy. Application to Europe and Eastern Turkey**

**2. Specific Remarks**

- Eq 1: Why did you not use the site term (e.g. Vs30 inferred) as an explanatory variable for the site effect (fixed effect), this helps reduce random variability; and the delats2 are used to consider uncertainties that are not taken into account by Vs30 (i.e. For example, if $VS_{30}$ is used for site classification, two sites with the same $VS_{30}$ can still have significantly different site profiles and therefore have different site amplifications).

- Eq 1 is a non-linear functional form; why did you use linear mixed effect model rather be non-linear model like INLA ?.

- Line 114 : rationalization ? do you mean regionalization.

- The use of a GMM model such as the one used in this study may complicate the interpretation of the results. I suggest you try a site amplification model: Amp(FAS_sur/FAS_Downhole) This way, you would only have delta_S2S  and delta_Amp (describes the record-to-record variability of the amplification at site s for earthquake e).

- To make sure there's a correlation between amplification factor (AF) and delta_S2_S, I'd like to have a figure that gives Amp vs exp(delta_S2S). Amp=FAS_soft_soil/FAS_Rock. You can use the EC8 classification.

- In figure 3, I wonder if the (non-Gaussian) distribution of geomorphological sediment thickness (e.g. more than 800 sites have H=0-2 m)... is this non-uniform distribution taken into account when building the model?.

- In eq 6, log is 't log10 or Ln ?. Also does Ys(f,Proxy) represent delta_S2S ?.

- I'm not convinced that delta_S2S can represent amplification itself. But rather the epistemic uncertainty of the site effect part (closely linked to the GMM used). In addition, amplification is normally unitless. However, here, delta_s2s takes unit of the FAS. Thank you for explaining this to me.

- Line 214 : extreme values  or outlier value  ?.

- Line 217 : What do you mean by "fold cross validation test". Give us some explanations. And why "10" fold ?.

- Figure 5, usually with  classical GMMs, we have Vref (e.g Vs30 = 760 m/s), here we don't see this threshold why ?.

- Eq 7 is nothing more and nothing less than the residual equation in equation 6. Why named correction term?. I would like to see a figure that gives the delta_S2Scor vs proxies for some frequencies, it gives us an idea on the presence or not of bias.

- You have chosen 0.46-9.9 Hz frequency range...I want to see the same curves as Figure 7, but for a wide range, e.g. 0.1 to 100 Hz.

- In line 304, you wrote: "The object of this study is to predict regional site amplification over a large area using regionally or globally available site proxies". In my view, this sentence must be in "introduction part".

- The comparison between Figure 8 (training phase) and Figure 3 (testing phase) are not consistent. In fact, you have to validate with a smaller interval (like 350-400 m/s) and add soft sites comparison.

**3. review conclusion**

This work cannot be published in its current state. I recommend to considering general and specific remarks. After that, the work can be published in the natural hazards and earth system sciences.

---

## Referee Comment (RC2)

Review comments for the paper entitled "Introducing inferred geomorphological sediment thickness as a new site proxy to predict ground-shaking amplification at regional scale. Application to Europe and Eastern Turkey » by Karina Loviknes et al.

Natural Hazards and Earth System Sciences

Manuscript Number: egusphere-2023-1370

The authors of this paper develops a new model based on the geomorphological sediment thickness (GST) derived from Pelletier et al. (2016) to predict site amplification at continental or regional scale. This new model is compared to three known models based respectively on Vs30 proxy, slope, and geological era. Then, the authors apply the four models at the border region between Turkey and Syria and, more locally, for three main cities of the region.

The proposed methodology is worth publishing, aligned with the scope of the natural hazards and earth system sciences. However, the objectives of the paper are not clear enough for the readership to endorse the authors's hypothesis. To my point of view, the rationale should be more detailed and the conclusions should emphasize the use limitations of the proposed model. Rephrasing should concern the points given below.

**The title of the paper** does not fully reflect the contents of the paper. In a large part of the paper, the authors compare the new model with three existing models. This comparison does not appear in the title.

**In the introduction (chapter 1)**, it is not clear why a new prediction model for site amplification is needed. A full discussion on the limits of the existing models (e.g. quality of input data, resolution, limits of application, etc…) and on the field of applications of this new model would probably better explain the choice of the authors. Would the new proposed model be applicable for a large number of GMM or only for GMM based on Kotha et al. method?

**In chapters 2 to 5**, some points should be clarified to make the text easy to understand to a diversified audience (including for non-specialists in GMM development):

- all the terms of the equations should be explained (for example Rref, a, g, Mh, etc…),
- the authors should explained the frequency values chosen for the tests (f=0.529, 1.062 and 9.903 Hz) yet the interpretation of the prediction results is frequency dependant and should be altered by the limitations of the selected model,
- l. 155: the authors explain that they have used a different processing from previous works: what is the impact of this choice?
- l. 158: the GST data does not extend beyond 50 m depth. What is the impact of this limitation on the prediction model, especially at low frequency?
- Paragraph 3.3: a discussion on the resolution of the geological era model would be necessary and its impact at high frequency,
- Paragraph 4:

- o which arguments could the authors present to demonstrate that the relation between site amplification and measured V30 is log-linear?
  - o Could the authors give estimators of the goodness of fit for all linear regressions?
  - o Could the authors improve their figures in terms of legibility (in particular dash lines are a poor graphical choice),
  - o l.236: high sediment thicknesses induce lower frequency site amplifications but not necessary higher site amplification than low sediment thicknesses,
  - o the geological era is inferred from a low resolution model: how this could impact
- Paragraph 5:
  - o Could the authors re-explicit the way the indicator works?
  - o Could the authors harmonize the symbology used for equations 7 and 8 and the symbology used for figure 7?
  - o The authors show that none of the amplification models are not distinguishable for frequencies above 3 Hz. What are the consequences of this statement? Does this mean that they cannot be used for frequencies above 3 Hz?

**In chapter 6**, the authors apply the new GST based model to Europe and to the Turkey-Syria border region. They compare the four models for three different soil classes based on Vs30 measured values (175 m/s, 375 m/s and 775 m/s). The main points to be discussed are the following:

- How does they choose those values? Are they representative of the distribution shown in Figure 3? In terms of site effects, the first and last class (175 and 775 m/s) correspond to very specific configurations (very soft soils and rocky sites): testing the model in more "regular" configurations should improve the robustness of the conclusions.
- l.310-314: I do not agree with the hypothesis associating soft soils to high GST values (and stiff soils with low GST values). This hypothesis does not take into account the Vs of the sedimentary layer. In the deep Tertiary basin for example, one could have a stiff soil (in terms of VS30) with high thicknesses and high site amplification. This point should be discussed in details and the impact of such hypothesis should be emphasize.
- Table 1: how many data are available in each site condition classes (soft soil/stiff soil/rock)?
- Figure 9: is it adequate to consider GST values inferior to 5 m (this configuration is considered as rock site in EC8 classification)? This point should be discussed.
- l 345-l. 351: the authors apply the four selected prediction models to Europe but they give neither spatial nor numerical indicator to compare the site amplification values of each model (for example through the plot of their respective distribution) and their respective impact on risk assesment. A thorough discussion on the results is necessary to emphasize the pros and cons of the proposed model. Though the aim of the paper is not to re-create the exact site-specific amplification, it should be also interesting to test whether the models are in adequation with the regional site amplification maps inferred from other site effects proxy (for example the Italian VS30 map of Mori et al).
- Figure 11: Where are the Taurus mountains? Please complete the maps with the countries names. At the Turkey-Syria border, the GST model presents a possible artefact and the geological era model does not provide data on the Syrian side. In those conditions, why did the authors choose this area for their test? Would not it have been more appropriate to use a region with a better coverage in terms of calibration data?
- Figure 12: Please complete with the February 2023 epicentres.
- l. 400 and more: As said by the authors, the proposed models are regional. In this context, is it correct to test them at city scale? If so, why did they choose the Antakya, Aleppo and Gaziantep

cities since they made no comparison with the observed site amplification or damage during the February 2023 events. What is the aim of this test?

**In the conclusion**, the first point is that the authors do not conclusively demonstrate the value of the proposed new model since it gives equivalent or weaker results than the other ones. The authors should give stronger arguments to show the interest of using their model in future hazard and risk studies at regional scale. A second point to address is that the application of the four models at European scale show a great variability. If the authors wish to continue applying these models, it is necessary to quantify the impact of such broad epistemic uncertainties for site amplification prediction on risk assessment. Third, the authors should not avoid discussing about the limitations of the proposed model (pros and cons) and should propose a plan of actions to improve it and, consequently, decrease the related uncertainties.

**Review conclusions:**

I recommend major revisions to strengthen the work done by the authors before publication.

---

## Author Comment (AC2)

We are grateful to the editor and the reviewers for their comments and suggestions and have made four main improvements to the manuscript:
1. We have elaborated on the description of the site-to-site term $\delta$S2S,

2. We have clarified that the resulting site amplification predictions should be interpreted in reference to the median prediction of the associated GMM,

3. We have simplified the section on Eastern Türkiye and included a comparison of the within-event residuals obtained using the GMM predictions and the four proxy-based site amplification models,

4. We have addressed all the minor comments and issues from the reviewers.

In the following we have addressed the comments and are describing the changes that have been done. The reviewer's comments are given in blue, and our replies in black.

**1. Reviewer Comments natural hazards and earth system sciences**

The authors have tested the geomorphological sediment thickness (GST) and classical local soil-conditions proxies such as Vs30 in response to this question: Is GST a relevant global site proxy for PSHA analysis. To do this, they used site-to-site residuals issued from Kotha FAS-Model.

The selected study areas and proposed methodology are of scientific and engineering interest, aligned with the scope of the natural hazards and earth system sciences. However, the claim that the developed model is based on site-to-site residuals using linear regression is questioned, the organization of the different sections can be improved, and the manuscript lacks adequate explanation in a few areas for a reviewer to fairly assess the technical quality of the study:

1. General remarks
- In my opinion, the use of delta_S2S residuals (which is considered here as an epistemic uncertainty) is strongly linked to the Kotha model. To validate the method, several GMMs need to be considered and it needs to be proven that the delta_S2S values do not depend too much on the GMM used.
   ◦ It is true that the $\delta$S2S$_s$ residuals are dependent on the GMM used and represent the systematic deviation of recorded ground motions from that GMM median predictions related to a site $s$. Because there is no proxy-based site term included in the GMM, the $\delta$S2S$_s$ residuals captures all the site-specific response, and can therefore be used to evaluate the ability site proxies to predict site amplification for that GMM. Section 2 have been renamed site-to-site term and partly rewritten to make this clearer (lines 88-96 and 117-121).

- For me, the simplest way is to deduce the amplification from the FAS ratio and to find a correlation between this ratio and the site proxies. e.g. if this ratio = 2 its interpretation is simple, but with a delta_S2S=-1.5 how can we interpret it especially this study targets ESHM20 by proposing a regional proxy (GST).?
   ◦ As described above, $\delta$S2S$_s$ is relative to the mean of all sites and not to a rock reference, meaning that $\delta$S2S$_s$ with a positive value are amplified compared to the mean, and a

negative value signifies de-amplification relative to the mean of all sites. Line 121 have been slightly rephrased to make this clearer.

- With this study we remark that Delta_S2S=f(Proxy) is an heteroscedastic model (Khota model). This Heteroscedasticity lead to biased estimates of the standard errors of the regression coefficients. This can make it difficult to determine the significance of the coefficients and can lead to incorrect conclusions about the relationship between the proxy in the model. Additionally, heteroscedasticity can also affect the efficiency of the parameter estimates,leading to less precise estimates than would be obtained with a homoscedastic model. I know you are looking for this heteroscedasticity, since the GMM (Kotha) model does not contain any proxy representing site effect. However, I'm wondering if, can you add a site proxy (e.g.GST) to the GMM model to ensure that the delta_S2S follows a homoscedastic model (goal is to validate GMM model).
  - ◦ This comment can be understood in two ways. Either it is indicated that heteroscedasticity may affect the results because the $\delta$S2S$_s$ may be magnitude dependent and include nonlinear amplification, or that the un-even distribution of the site proxies may bias the resulting site amplification models.
  - ◦ Regarding the first issue, because $\delta$S2S$_s$ can be derived on a limited number of events (minimum 3), magnitude bias and nonlinearity may affect the $\delta$S2S$_s$. However, as also argued by Weatherill et al., 2023, $\delta$S2S$_s$ is generally assumed to be linear because most dataset contains mainly lower magnitude intensity ground motions, which is also the case for the ESM dataset which has few records from large earthquakes on soft soils stations, this has also been observed by Guéguen et al. (2019)
  - ◦ For the second issue, the following fitted (predicted) model vs residuals figure show that although the residuals of the site amplification models have a large scatter, they are more or less equally distributed along the fitted values (x-axes) range, except for a few outliers and a slight downgoing trend at high predicted values based on slope. We therefore assume that site amplification models are not demonstratively heteroscedastic:

[Figure]

**Figure 1 fitted (predicted) model vs residuals between δS2S$_s$ and the site amplification models based on inferred V$_{S30}$ (top row) slope (second top row), geomorphological sedimentary thickness (second bottom row) and geological era and slope (bottom row), for f = 0.529 Hz (left), f = 1.062 Hz and f = 9.903 Hz.**

- Why did you use linear regression to have Site amplification factor ?. Site behavior is so complicated that a simple linear model is insufficient to consider the underlying physics. Or use small strain conditions to remove nonlinear effect. In addition, delta_S2S. In this sense, it must be shown that the model developed does not suffer from underfitting.
  - It is indeed true that site behavior is too complex for a linear relation to fully capture, but because the main object of this study is to compare the ability of the proxies to predict site amplifications, and not to find the best possible relation between the site amplification and the proxies, we use linear regression for the simplicity. This has been clarified in lines 272-274.

- And I wonder how to obtain the amplification value with delta_S2S.
  - As described above, because no proxy-based site term is included in the GMM used to derive the $\delta S2S_s$., $\delta S2S_s$. captures all the site-specific response, and can therefore be used as an empirical site-amplification function describing the local amplification, or de-amplification, of each station with respect to the median of all sites. Section 2 have been renamed site-to-site term and have been partly rewritten to make this clearer (lines 88-96 and 117-121).

- Finally, if this is a sensitivity study, it should be mentioned in the title. Like : Using site-to-site residuals to testing the Relevance of geomorphological sediment thickness as a regional site proxy. Application to Europe and Eastern Turkey
  - We have changed the title to "Exploring inferred geomorphological sediment thickness as a new site proxy to predict ground-shaking amplification at regional scale. Application to Europe and Eastern Turkey", and lines 78-80 have been rephrased to make it clearer that we are using site-to-site terms.

2. Specific Remarks
- Eq 1: Why did you not use the site term (e.g. Vs30 inferred) as an explanatory variable for the site effect (fixed effect), this helps reduce random variability; and the delats2 are used to consider uncertainties that are not taken into account by Vs30 (i.e. For example, if VS30 is used for site classification, two sites with the same VS30 can still have significantly different site profiles and therefore have different site amplifications).
  - As described above, using no proxy-based, like VS30, site term in GMM makes sure the resulting $\delta S2S_s$ residuals capture all the site-specific response, this has been better emphasized in lines 116-119.

- Eq 1 is a non-linear functional form; why did you use linear mixed effect model rather be non-linear model like INLA ?.
  - We use the robust linear mixed-effects regression to derive the random effects while down-weighting the outliers, and to stay consistent with previous work. However, using INLA, which uses a Bayesian framework, might be a possibility in future work.

- Line 114 : rationalization ? do you mean regionalization.
  - Yes, thank you for pointing out this error, it has been corrected.

- The use of a GMM model such as the one used in this study may complicate the interpretation of the results. I suggest you try a site amplification model:

Amp(FAS_sur/FAS_Downhole) This way, you would only have delta_S2S and delta_Amp (describes the record-to-record variability of the amplification at site s for earthquake e).

- ◦ Because we are looking at Europe-wide amplification, deriving the amplification from the spectral ratio with a nearby rock reference or borehole station is not an alternative, a paragraph in lines 89-96 have been added to make this clearer.

- • To make sure there's a correlation between amplification factor (AF) and delta_S2_S, I'd like to have a figure that gives Amp vs exp(delta_S2S). Amp=FAS_soft_soil/FAS_Rock. You can use the EC8 classification.
  - ◦ As described above, using spectral ratio with nearby rock reference is not an alternative when wanting to obtain amplification factors for such a large area as Europe, lines 89-96 have been added to make this clearer. However, $\delta$S2S$_s$ can be used as an empirical site-amplification function describing the local amplification, or de-amplification, of each station with respect to the median of all sites (Kotha et al., 2018). In fact, previous studies have compared $\delta$S2S$_s$ to and show strong similarities with amplification factors derived using a generalized inversion technique (GIT, e.g. Bindi et al., 2017; Wang et al., 2023), this is also described in line 122:

[Figure]

**Figure2: from Wang et al. (2022), comparing horizontal site amplification factors (HSAF) estimated using GMM for the Fourier spectra performed by Loviknes et al. (2021) Hybrid method 2, and GIT at the Japanese Kik-net stations (A) IBUH01, (B) KGWH03, (C) KGWH04, (D) SBSH09, and (E) TTRH07.**

- • In figure 3, I wonder if the (non-Gaussian) distribution of geomorphological sediment thickness (e.g. more than 800 sites have H=0-2 m)... is this non-uniform distribution taken into account when building the model?.
  - ◦ Yes, as discussed in lines, the non-Gaussian distribution of GST is caused by the extreme/end-values 0 and 50 m. To deal with this issue we have attempted censored regression, removing the extreme values and are finally using 10-fold validation to ensure that the linear regression is not to depended on the station distribution. Line 243 have been rephrased to make this clearer.

- In eq 6, log is 't log10 or Ln ?. Also does Ys(f,Proxy) represent delta_S2S ?.
  - Yes it should be ln, thank you for pointing this out. Ys(f,Proxy) represent the predicted $\delta S2S_s$ based on the proxy-based site amplification model derived from the linear regression, is has been better clarified in line 211.

- **I'm not convinced that delta_S2S can represent amplification itself.** But rather the epistemic uncertainty of the site effect part (closely linked to the GMM used). In addition, amplification is normally unitless. However, here, delta_s2s takes unit of the FAS. Thank you for explaining this to me.
  - As described above, $\delta S2S_s$ residuals represent the amplification, or de-amplification, of each station with reference to the median GMM prediction for all the sites. $\delta S2S$ is unitless and $\exp(\delta S2S_s)$ is the amplification factor. However $\delta S2S$ is derived for FAS (Fourier amplitude spectra) and not for response spectra (which is the common practice).

- Line 214 : extreme values or outlier value ?.
  - Extreme values here refer to the values on each end of the range, so for GST, 0 and 50 m, to avoid confusion the word extreme has been changed to end-value in line 259.

- Line 217 : What do you mean by "fold cross validation test". Give us some explanations. And why "10" fold ?.
  - The explanation of the method has been improved in lines 264-267.

- Figure 5, usually with classical GMMs, we have Vref (e.g Vs30 = 760 m/s), here we don't see this threshold why ?.
  - As described above, no VS30-term is included in the GMM and a refernce VS30 is therefore not used. Instead $\delta S2S_s$ is referenced to the median of all the sites, not to rock sites.

- Eq 7 is nothing more and nothing less than the residual equation in equation 6. Why named correction term?. I would like to see a figure that gives the delta_S2Scor vs proxies for some frequencies, it gives us an idea on the presence or not of bias.
  - The correction term $\delta S2S_{s,cor}(f)$ in equation 7 and 8 represents the remaining site amplification that is not captured by the proxy based amplification prediction $\delta S2S_s(f,Proxy)$ and $\Phi_{s2s,\ cor.}(f,Proxy)$ is the site-to-site variability of $\delta S2S_{s,cor}(f)$. If the site amplification model where able to perfectly predict and capture the full range of the site amplification at a specific site, $\varphi_{s2s,\ cor.}(f,Proxy)$ would be reduced to zero. However, such an ideal case is not realistic and conventional site amplification models can only aim to reduce the $\Phi_{s2s,\ cor.}(f,Proxy)$ as much as possible. A lower $\Phi_{s2s,\ cor.}(f,Proxy)$, therefore indicates that the proxy are able to capture the site amplification. This has been clarified in lines 322-316. The new figure A3 showing $\delta S2S_{s,cor}(f)$ with the proxies are included in the appendix, and show that there are obvious biases as $\delta S2S_{s,cor}(f)$ is evenly distributed around zero.

- In line 304, you wrote: "The object of this study is to predict regional site amplification over a large area using regionally or globally available site proxies". In my view, this sentence must be in "introduction part".
  - This sentence has been rephrased to "The object of this study is to test regionally or globally available site proxies as predictors for regional site amplification over a large

area" (now line 369) and parts of the introduction have been rephrased to make the object clearer.

- The comparison between Figure 8 (training phase) and Figure 3 (testing phase) are not consistent. In fact, you have to validate with a smaller interval (like 350-400 m/s) and add soft sites comparison.
  ◦ Thank you for pointing this out, we have changed Figure 8 (now Fig. 11) to follow the Eurocode 8 classes instead of the Vs30 intervals to better justify the selected ranges and follow the distribution in Figure 3.

**References**

Bindi, D., Spallarossa, D., and Pacor, F.: Between-event and between-station variability observed in the Fourier and response spectra domains: comparison with seismological models, Geophysical Journal International, 210, 1092–1104, https://doi.org/10.1093/gji/ggx217, 2017.

Guégen P, Bonilla F, Douglas J: Comparison of soil nonlinearity (In Situ Stress-Strain Relation and G/Gmax Reduction) observed in strong-motion databases and modeled in ground-motion prediction equations. Bull Seismol Soc Am 109(1):178–186, 2019.

Kotha, S. R., Cotton, F., and Bindi, D.: A new approach to site classification: mixed-effects ground motion prediction equation with spectral clustering of site amplification functions, Soil Dynamics and Earthquake Engineering, 110, 318–329, https://doi.org/10.1016/j.soildyn.2018.01.051, 2018.

Loviknes, K., Kotha, S. R., Cotton, F., and Schorlemmer, D.: Testing nonlinear amplification factors of ground-motion models, Bulletin of the Seismological Society of America, 111, 2121–2137, https://doi.org/10.1785/0120200386, 2021.

Wang, Z., Nakano, K., Ito, E., Kawase, H., and Matsushima, S.: A hybrid approach for deriving horizontal site amplification factors considering both the similarity of HVSRe and the vertical amplification correction function, Earthquake Engineering & Structural Dynamics, 52, 128–146, https://doi.org/10.1007/s10518-023-01712-z, 2023.

Weatherill, G., Crowley, H., Roullé, A., Tourlière, B., Lemoine, A., Gracianne, C., Kotha, S. R., and Cotton, F.: Modelling site response at regional scale for the 2020 European Seismic Risk Model (ESRM20), Bulletin of Earthquake Engineering, 21, 665–714, https://doi.org/10.1007/s10518-022-01526-5, 2023.

---

## Author Comment (AC3)

We are grateful to the editor and the reviewers for their comments and suggestions and have made four main improvements to the manuscript:

1. We have elaborated on the description of the site-to-site term $\delta$S2S,

2. We have clarified that the resulting site amplification predictions should be interpreted in reference to the median prediction of the associated GMM,

3. We have simplified the section on Eastern Türkiye and included a comparison of the within-event residuals obtained using the GMM predictions and the four proxy-based site amplification models,

4. We have addressed all the minor comments and issues from the reviewers.

In the following we have addressed the comments and are describing the changes that have been done. The reviewer's comments are given in blue, and our replies in black.

**2. Review comments for the paper entitled "Introducing inferred geomorphological sediment thickness as a new site proxy to predict ground-shaking amplification at regional scale. Application to Europe and Eastern Turkey » by Karina Loviknes et al. Natural Hazards and Earth System Sciences**

The authors of this paper develops a new model based on the geomorphological sediment thickness (GST) derived from Pelletier et al. (2016) to predict site amplification at continental or regional scale. This new model is compared to three known models based respectively on Vs30 proxy, slope, and geological era. Then, the authors apply the four models at the border region between Turkey and Syria and, more locally, for three main cities of the region. The proposed methodology is worth publishing, aligned with the scope of the natural hazards and earth system sciences.

However, the objectives of the paper are not clear enough for the readership to endorse the authors's hypothesis. To my point of view, the rationale should be more detailed and the conclusions should emphasize the use limitations of the proposed model. Rephrasing should concern the points given below.

- **The title of the paper** does not fully reflect the contents of the paper. In a large part of the paper, the authors compare the new model with three existing models. This comparison does not appear in the title.
  ◦ We have changed the title to "Exploring inferred geomorphological sediment thickness as a new site proxy to predict ground-shaking amplification at regional scale. Application to Europe and Eastern Turkey", and also stressed the comparison more in the introduction (e.g. line 65)

- In the introduction (chapter 1), it is not clear why a new prediction model for site amplification is needed. A full discussion on the limits of the existing models (e.g. quality of input data, resolution, limits of application, etc…) and on the field of applications of this new model would probably better explain the choice of the authors. Would the new proposed model be applicable for a large number of GMM or only for GMM based on Kotha et al. Method?

- ◦ The resulting site amplification predictions and maps should only be interpreted in reference to the associated GMM used to derive $\delta S2S_s$, this has been clarified in the caption of Figures 12, 14, A5 and A6 and in the text (e.g. lines 275-276 and 442)
- ◦ The aim of this study, and of deriving the site amplification models, is to test whether GST add any valuable information to site amplification prediction and can be used as a site proxy. To answer this question, we derive site amplification model using both new and common proxies and compare the ability of the site proxies to predict the observed amplification. The main aim is therefore not to launch a new site amplification model, but to examine the effects of using different site proxies as alternatives to VS30. The limitations of VS30 are stated in the introduction. In order to clarify this, we have rephrased some of the descriptions of the object of the study, for example in line 72.

- In chapters 2 to 5, some points should be clarified to make the text easy to understand to a diversified audience (including for non-specialists in GMM development):
  - ◦ all the terms of the equations should be explained (for example Rref, a, g, Mh, etc…),
    - ▪ Thank you for pointing this out, we have added a sentence explaining the reference values in lines 113-114.

  - ◦ the authors should explained the frequency values chosen for the tests (f=0.529, 1.062 and 9.903 Hz) yet the interpretation of the prediction results is frequency dependant and should be altered by the limitations of the selected model,
    - ▪ The three frequency values are selected to show the amplification at three different frequency values, the test is however performed at all frequencies between $f = 0.460 - 9.903$ Hz, this has been clarified in line 138.

  - ◦ l. 155: the authors explain that they have used a different processing from previous works: what is the impact of this choice?
    - ▪ The difference from previous work is not directly the processing applied, but the value used. The Pelletier et al. (2016) provided several different grids describing different aspects of regolith, soil and sediment thickness. In this study we use the combination of the two soil and sediment thickness grids (for hillslope and upland/lowland) in order to access the value for a broader area. Several clarifications have been made in lines 170-180.

  - ◦ l. 158: the GST data does not extend beyond 50 m depth. What is the impact of this limitation on the prediction model, especially at low frequency?
    - ▪ At low frequency the performance of the site amplification model based on GST is slightly reduced, this is now discussed in lines 252-255.

  - ◦ Paragraph 3.3: a discussion on the resolution of the geological era model would be necessary and its impact at high frequency,
    - ▪ The resolution of the geological era (1:1,500,000) have been added in lines 188 and 194 and is also attributed as passible factor to explain the poor performance of the models at high frequencies.

- Paragraph 4:
  - ◦ which arguments could the authors present to demonstrate that the relation between site amplification and measured V30 is log-linear?

- We have added a figure (new Fig. 4) showing that the linear regression between ds2s and ln(VS30) have a higher correlation coefficient than the linear regression between ds2s and VS30 in linear scale, this is also described in lines 216-222.

- Could the authors give estimators of the goodness of fit for all linear regressions?
  - Yes, thank you for pointing out that this was missing. We have added a figure (new Fig. 6) showing the coefficient of determination for the different regressions shown in Figure 5, and a figure in the Appendix (Fig. A2) showing the coefficient of determination for the linear regressions shown in Figure 5 and 6.

- Could the authors improve their figures in terms of legibility (in particular dash lines are a poor graphical choice),
  - We have altered some of the Fig 4 (new Fig 5) and color scales used for the lines, but in some cases dashed lines are still used in order to keep the figures colorblind friendly.

- l.236: high sediment thicknesses induce lower frequency site amplifications but not necessary higher site amplification than low sediment thicknesses,
  - Yes, but the sentence in line 236 (now line 289), only refers to the general trends observed in Fig. 7 and are not trying to make any conclusions about site-specific amplification.

- the geological era is inferred from a low resolution model: how this could impact
  - As described above, the resolution of the geological era could be one of the causes for the poor performance of the models at high frequencies, this is mentioned in line 337.

- Paragraph 5:
  - Could the authors re-explicit the way the indicator works?
    - Yes, we have added lines 311-316 to better explain how the indicators work.

  - Could the authors harmonize the symbology used for equations 7 and 8 and the symbology used for figure 7?
    - Yes, thank you for pointing this out, style of phi has been changed.

  - The authors show that none of the amplification models are not distinguishable for frequencies above 3 Hz. What are the consequences of this statement? Does this mean that they cannot be used for frequencies above 3 Hz?
    - Yes, and a similar result was also found by Bergamo et al. (2022) who also used higher resolution site proxies. Bergamo et al. (2022) found that low resolution proxies worked best at low frequencies, higher resolution proxies worked well at intermediate frequencies, while direct proxies work at a wider frequency range (up to 5Hz). Lines 338-341 have been added to emphasize this.

- In chapter 6, the authors apply the new GST based model to Europe and to the Turkey-Syria border region. They compare the four models for three different soil classes based on Vs30 measured values (175 m/s, 375 m/s and 775 m/s). The main points to be discussed are the following:
  - How does they choose those values? Are they representative of the distribution shown in Figure 3? In terms of site effects, the first and last class (175 and 775 m/s) correspond to

very specific configurations (very soft soils and rocky sites): testing the model in more "regular" configurations should improve the robustness of the conclusions.

- Thank you for this comment, the three ranges (soft soil, soil and rock) were chosen to show how the model perform for very different soil types, but it is true that these ranges do not represent well the distribution in figure 3. We have therefore changed figure 8 (now Fig. 11) to follow the Eurocode 8 (EC8) classes instead of Vs30 intervals, both to better justify the choice of values and to better follow the distribution in Figure 3.

- l.310-314: I do not agree with the hypothesis associating soft soils to high GST values (and stiff soils with low GST values). This hypothesis does not take into account the Vs of the sedimentary layer. In the deep Tertiary basin for example, one could have a stiff soil (in terms of VS30) with high thicknesses and high site amplification. This point should be discussed in details and the impact of such hypothesis should be emphasize.

  - As described in the comment above, Figure 8 (now Fig. 11) have been changed to follow the EC8 classes instead of Vs30 and the GST values are now chosen based on the new draft for EC8 following Paolucci et al. (2021)

- Table 1: how many data are available in each site condition classes (soft soil/stiff soil/rock)?

  - The value has been added and Table 1 have been changed to match the EC8 classifications as shown in new Fig 11.

- Figure 9: is it adequate to consider GST values inferior to 5 m (this configuration is considered as rock site in EC8 classification)? This point should be discussed.

  - As described in the comments above, the GST values are now selected based on the new draft for EC8 following Paolucci et al. (2021)

- l 345-l. 351: the authors apply the four selected prediction models to Europe but they give neither spatial nor numerical indicator to compare the site amplification values of each model (for example through the plot of their respective distribution) and their respective impact on risk assesment. A thorough discussion on the results is necessary to emphasize the pros and cons of the proposed model. Though the aim of the paper is not to re-create the exact site- specific amplification, it should be also interesting to test whether the models are in adequation with the regional site amplification maps inferred from other site effects proxy (for example the Italian VS30 map of Mori et al).

  - As described above, the object of this study, and of deriving the site amplification models, is to assess the ability of the proxies to predict site amplification, not to develop and propose new site amplification model. Several sentences have been rephrased to make this clearer (e.g. line 367-368 and 525).
  - The results of this study have shown that using different site proxies to predict site amplification gives notable different results and captures the epistemic uncertainty associated with modelling site amplification when using inferred proxies. This epistemic uncertainty should be incorporated into the final risk calculation, however, performing risk and loss calculations using the proxy-based site amplification models is beyond the scope of this work. A discussion on the impact the study has on risk assessment is added in paragraph in lines 535-544.

- Figure 11: Where are the Taurus mountains? Please complete the maps with the countries names. At the Turkey-Syria border, the GST model presents a possible artefact and the geological era model does not provide data on the Syrian side. In those conditions, why did the authors choose this area for their test? Would not it have been more appropriate to use a region with a better coverage in terms of calibration data

  The area was chosen because the recent earthquakes demonstrated the need for cross-boundary site amplification maps. In addition, a comparison using the ground motions recorded during the recent events has been added (lines 491-510).

- Figure 12: Please complete with the February 2023 epicentres.
  - This has been done in Fig 11 (now Fig. 13) and the new Figure 15 showing the distribution of the new dataset.

- l. 400 and more: As said by the authors, the proposed models are regional. In this context, is it correct to test them at city scale? If so, why did they choose the Antakya, Aleppo and Gaziantep cities since they made no comparison with the observed site amplification or damage during the February 2023 events. What is the aim of this test?
  - While it is true that the site proxies and models are only meant for regional scale, the city comparison serves to emphasize how different the predictions are and the importance of capturing epistemic uncertainty when adopting these approaches in seismic risk analysis or even in rapid post-event impact assessment. However, as also stated in the manuscript, this comparison is not entirely appropriate, and we realize that it might also add to the confusion about of the aim of the models. We have therefore removed this part and instead added a comparison of the models to the within-event residuals from the recent events (lines 491-510).

- In the conclusion, the first point is that the authors do not conclusively demonstrate the value of the proposed new model since it gives equivalent or weaker results than the other ones. **The authors should give stronger arguments to show the interest of using their model in future hazard and risk studies at regional scale.** A second point to address is that the application of the four models at European scale show a great variability. **If the authors wish to continue applying these models, it is necessary to quantify the impact of such broad epistemic uncertainties for site amplification prediction on risk assessment.** Third, the authors should not avoid discussing about the limitations of the proposed model (pros and cons) and should propose a plan of actions to improve it and, consequently, decrease the related uncertainties.
  - As described above, the object of this study is to evaluate the ability of the proxies to predict site amplification, the models are therefore exploratory and not developed with the aim of capturing the best possible relation between the proxies and empirical amplification. Several sentences have been rephrased to make this clearer (e.g. line 368-369 and 524).
  - Furthermore, the results of this study have shown that using different site proxies to predict site amplification gives significantly different results. This emphasizes the importance of capturing the epistemic uncertainty associated with modelling site amplification when using inferred proxies. The epistemic uncertainty needs to be incorporated into the final risk calculation and to fully assess the impact of this epistemic uncertainty, risk and loss calculations should be performed using the different site amplification models, however, this is beyond the scope of this work. This is discussed in the added paragraph in lines 536-544 and emphasized more in the conclusion.

**References**

Bergamo, P., Hammer, C., and Fäh, D.: Correspondence between Site Amplification and Topographical, Geological Parameters: Collation of Data from Swiss and Japanese Stations, and Neural Networks-Based Prediction of Local Response, Bulletin of the Seismological Society of America, 112, 1008–1030, https://doi.org/10.1785/0120210225, 2022.

Pelletier, J. D., Broxton, P. D., Hazenberg, P., Zeng, X., Troch, P. A., Niu, G.-Y., Williams, Z., Brunke, M. A., and Gochis, D.: A gridded global data set of soil, intact regolith, and sedimentary deposit thicknesses for regional and global land surface modeling, Journal of Advances in Modeling Earth Systems, 8, 41–65, https://doi.org/10.1002/2015MS000526, 2016.

Paolucci, R., Aimar, M., Ciancimino, A., Dotti, M., Foti, S., Lanzano, G., Mattevi, P., Pacor, F., and Vanini, M.: Checking the site categorization criteria and amplification factors of the 2021 draft of Eurocode 8 Part 1–1, Bulletin of Earthquake Engineering, 19, 4199–4234, https://doi.org/10.1007/s10518-021-01118-9, 2021.